# SSDCN: Spatial-Spectral Dual-Clustering-based Network for Hyperspectral Image Super-resolution

Yong Yang [1]   Xuran Zhang [1]   Shuying Huang [1]   Xiaozheng Wang [1]   Weiguo Wan [2]   Hangyuan Lu [3]

## Abstract

Hyperspectral image single image super resolution (HSI-SISR) faces a conflict between computational efficiency and global non-local modeling. Existing Transformers suffer from quadratic complexity, while window-based methods compromise global capture. To address this, we propose the spatial-spectral dual-clustering-based network (SSDCN). Our method introduces three innovations. First, we design a spatial-spectral dual-cluster block (SSDCB). Replacing expensive point-to-point attention, it uses content-driven clustering to learn low-rank structural bases, achieving global modeling with linear complexity $\mathcal{O}(KN)$. Second, we propose a pyramid progressive hierarchical architecture with a feature reuse reconstruction block (FRRB). It reuses the core tensor and spectral factors from coarse levels, updating only spatial factors to minimize redundancy. Third, we propose a pyramid hierarchical reconstruction joint loss to supervise intermediate levels, ensuring structural accuracy and preventing error accumulation. Experiments demonstrate that SSDCN surpasses SOTA methods in metrics and visual quality with significantly fewer parameters and FLOPs, achieving an optimal efficiency-performance balance.

## 1. Introduction

Hyperspectral images (HSIs), with their rich spectral signatures, are pivotal in remote sensing but often suffer from low spatial resolution due to sensor hardware constraints (Huang et al., 2024). To overcome this, single image super-resolution (SISR) has emerged as a critical research direction. Existing approaches generally fall into two categories: model-based and deep learning-based methods. Model-based methods, which leverage priors such as sparse representation (Gkillas et al., 2023; Xu et al., 2019) and low-rank tensor decomposition (Khader et al., 2022), offer strong physical interpretability (Huang et al., 2022). However, they are inherently limited by complex iterative optimization processes, slow inference speeds, and a struggle to model complex non-linear degradations. Conversely, deep learning-based methods, particularly 3D convolutional neural networks (CNNs), have achieved significant performance gains (Li et al., 2020; 2021). Yet, they face a fundamental trade-off between efficiency and performance: 3D convolutions incur high computational costs, while their restricted local receptive fields fail to capture the long-range spatial-spectral dependencies essential for high-quality reconstruction.

Recently, Transformers have been introduced to address these limitations by utilizing global attention mechanisms. However, the quadratic computational complexity $\mathcal{O}(N^2)$ of standard self-attention imposes an excessive burden on high-dimensional HSIs. To mitigate this, mainstream methods have adopted approximation strategies, but these come with distinct drawbacks. Window-based attention methods (e.g., CST (Chen et al., 2024), LDERT (Li et al., 2025b),SSATDP (Zhang et al., 2025b) ,WWDCST (Zhang et al., 2025a)) partition the image into fixed regions, which severs global context and relies on inefficient shift/shuffle operations for information propagation, leading to lagged pixel interactions. Alternatively, Spectral-wise attention methods (e.g., MSDFormer (Chen et al., 2023)) compute attention solely along the spectral dimension. While computationally lighter, this approach neglects spatial non-local self-similarity, often requiring auxiliary modules to compensate for missing spatial dependencies. Furthermore, directly processing hundreds of redundant spectral bands without effective dimensionality reduction significantly increases computational load and risk of distortion.

To resolve these challenges and fully exploit global low-rank priors, we propose the spatial-spectral dual-clustering-based network (SSDCN). Our core motivation is that global HSI correlations should be modeled not by expensive point-to-

[1]Tiangong University, Tianjin, China [2]Jiangxi University of Finance and Economics, Nanchang, China [3]Jinhua University of Vocational Technology, Jinhua, China. Correspondence to: Shuying Huang <huangshuying@tiangong.edu.cn>.

*Proceedings of the 43rd International Conference on Machine Learning*, Seoul, South Korea. PMLR 306, 2026. Copyright 2026 by the author(s).

point attention, but by efficiently learning and reorganizing intrinsic low-rank structural bases. We introduce the spatial-spectral dual-cluster block (SSDCB). This module performs successive clustering in both spatial and spectral dimensions to reduce dimensionality. Specifically, pixels and spectral bands interact with a small set of cluster centers to generate factor matrices. This content-aware mechanism achieves global modeling with linear complexity $\mathcal{O}(KN)$, effectively extracting high-information features while suppressing redundancy.

Given the cross-scale consistency of HSI spectral characteristics, we construct a pyramid progressive architecture and introduce a feature reuse reconstruction block (FRRB). Leveraging the hierarchical invariance of the core tensor and spectral factor matrix, the FRRB directly reuses $\times 1$ level features in subsequent reconstruction stages (e.g., $\times 2$, $\times 4$) and dynamically updates only the spatial factor matrix, thereby avoiding repetitive spectral decomposition calculations. Meanwhile, we propose a pyramid hierarchical reconstruction joint loss. By applying explicit supervision to each level, this mechanism ensures the physical fidelity of intermediate features, effectively curbing error accumulation in multi-stage networks and enhancing training stability.

In summary, the main contributions of this paper are as follows:

- We propose the SSDCB. It captures global spatial-spectral correlations with linear complexity $\mathcal{O}(KN)$, overcoming the computational bottleneck of traditional Transformers.

- We construct a pyramid hierarchical reconstruction architecture. Leveraging the hierarchical consistency of HSIs, we introduce the FRRB that reuses the core tensor and spectral factors, dedicating computation solely to the dynamic update of spatial factors, thereby minimizing redundancy.

- We design a pyramid hierarchical reconstruction joint loss to explicitly constrain intermediate outputs, ensuring the precision of the extracted core tensor and enhancing training stability across scales.

- Extensive experiments on multiple datasets demonstrate that SSDCN achieves SOTA performance while maintaining significantly fewer parameters and FLOPs compared to existing Transformer-based models.

## 2. Related work

### 2.1. Deep Learning-based Hyperspectral Image Super-Resolution

Deep learning-based HSI-SISR research aims to effectively capture the intrinsic spatial non-local self-similarity and spectral low-rank priors of HSI. Early approaches like GDRNN (Li et al., 2018)and EUnet(Liu et al., 2023) utilized 2D convolutions or unfolding networks, which were efficient but often lacked sufficient exploration of spectral information. To better exploit spatial-spectral dependencies, 3DFCNN introduced 3D convolutions(Sánchez-Caballero et al., 2022). Subsequent hybrid methods such as MCNet(Li et al., 2020) and ERCSR (Li et al., 2021)combined 2D and 3D convolutions to balance performance and parameter efficiency, while ReS$^3$-ConvSet(Hou et al., 2024) employed 1D kernels to further reduce model complexity. However, CNN-based methods remain limited by fixed local receptive fields and the high computational cost of 3D operations.

Recently, Transformers have been introduced to overcome these limitations with their powerful global modeling capabilities. To mitigate the quadratic complexity of standard self-attention, various efficient approximations have been developed. Spectral-wise attention methods, such as MSDFormer (Chen et al., 2023)and ESSAFormer (Zhang et al., 2023), focus on modeling spectral correlations to reduce spatial computation. Meanwhile, window-based approaches like CST (Chen et al., 2024) and WWDCST (Zhang et al., 2025a) utilize cross-scope aggregation and wavelet enhancements to capture features within restricted windows. LDERT (Li et al., 2025b) further advanced this by introducing multi-shape spatial rectangle attention. Despite these improvements, existing Transformer variants often rely on local or window-based approximations, which restricts their ability to fully leverage global non-local self-similarity and low-rank properties.

### 2.2. Dimensionality Reduction Techniques in Hyperspectral Images

HSI contain hundreds of continuous narrow bands, resulting in high spectral redundancy and the "curse of dimensionality," which introduces noise and burdens computation. Therefore, effective dimensionality reduction is a critical research area. Band selection aims to choose an informative subset of bands. Traditional methods like DHCA (Ji et al., 2022) employ divisive hierarchical clustering with local density to robustly capture spectral hierarchy and suppress noise. Similarly,ASPS (Wang et al., 2019) utilizes an adaptive subspace partition strategy with a coarse-to-fine approach to achieve efficient clustering.

In the deep learning domain, this concept has been adapted for super-resolution. SCPSN (Yang et al., 2024) proposes a spectral clustering block (SCB) that uses K-Means on the correlation matrix to select representative "super channels," significantly reducing network complexity. Another approach, LDERT (Li et al., 2025b), exploits HSI's low-rank properties via latent diffusion models to generate image-specific latent dictionaries. However, the use of diffusion

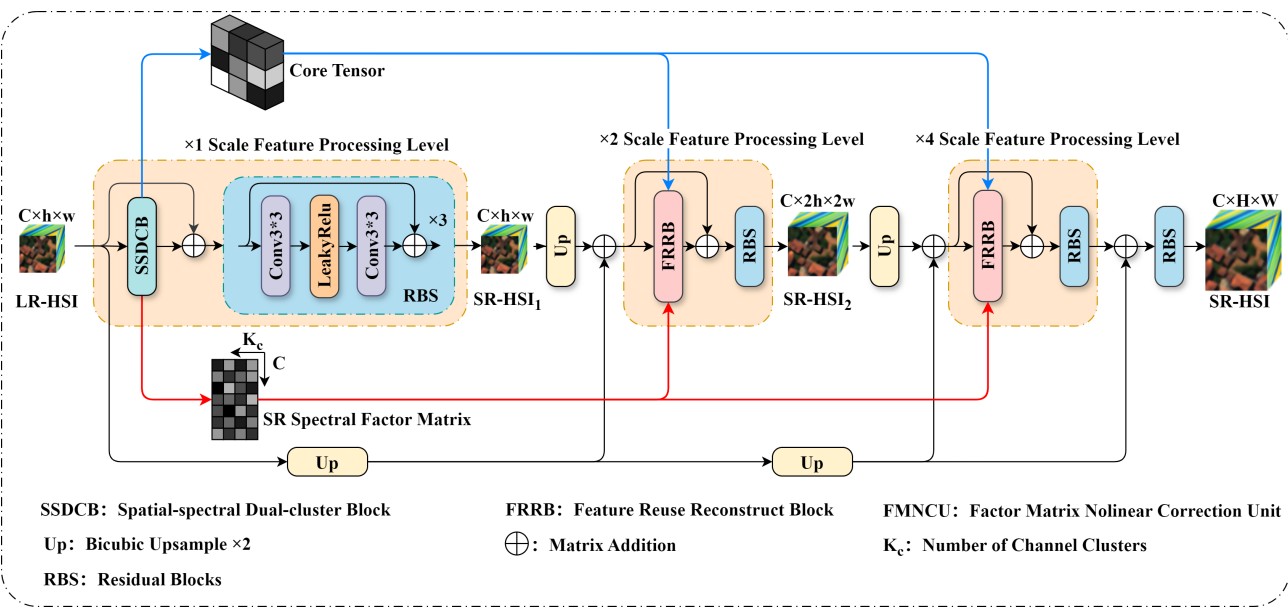

*Figure 1.* The overall architecture of the proposed Spatial-spectral Dual-clustering-based Network (SSDCN) for ×4 scale task.

models compromises inference speed. In contrast, this paper employs a spatial-spectral dual clustering method to perform dimensionality reduction in both dimensions, achieving an excellent balance between performance and inference speed.

## 3. Method

### 3.1. Overall Architecture of the Model

As shown in Figure 1, SSDCN adopts a coarse-to-fine pyramid progressive architecture, gradually recovering high-frequency details through concatenated feature levels. For regular scales that are powers of 2 (such as $4\times$), the network sequentially contains $\times1$, $\times2$, and $\times4$ processing stages, with levels connected by $\times2$ Bicubic upsampling.

In terms of module allocation, the $\times1$ level uses SSDCB to extract baseline features, while subsequent levels uniformly use FRRB for feature reuse and reconstruction. For a scaling factor $s$ that is not a power of 2, the model strategically builds to the nearest power of 2 level $s_{near}$ that is smaller than $s$, and dynamically adjusts the scaling factor of the last upsampling module to $s/s_{near}$ to achieve precise adaptation to any scale.

$I_{SR} \in \mathbb{R}^{C \times H \times W}$ represents the output image, where $H = s \times h$ and $W = s \times w$. $H_{level\times x}$ represents the processing method of the $x$ scale level. $H_{up\times s}$ represents the processing method of Bicubic upsampling by $s$ times. $H_{RBS}$ represents a processing module composed of three concatenated residual blocks, where each residual block consists of two convolutional layers, a LeakyRelu activa-

tion function, and a global residual skip connection. Then SSDCN can be expressed as follows:

$$
\begin{aligned}
I_{SR} &= H_{SSDCN\times s}(I_{LR}) \\
&= H_{RBS}\Big(H_{level\times s}\big(\ldots H_{up\times 2}\big(H_{level\times 2} \\
&\quad \big(H_{up\times 2}(H_{level\times 1}(I_{LR})) + I_{LR}\big)\big) \\
&\quad + H_{up\times 2}(I_{LR})\ldots\big) + H_{up\times s}(I_{LR})\Big),
\end{aligned}
\tag{1}
$$

where $H_{level\times 1}(I) = H_{RBS}(H_{SSDCB}(I) + I)$, and $H_{level\times s}(I) = H_{RBS}(H_{FRRB}(I) + I), s \neq 1$, $I_{LR} \in \mathbb{R}^{C \times h \times w}$ represents the input image, $I$ is a formal parameter.

### 3.2. Spatial-spectral Dual-cluster Block (SSDCB)

**Spectral Clustering and Spectral Factor Matrix**: Hyperspectral image bands are numerous and adjacent bands have extremely high redundancy. In order to achieve efficient modeling in the spectral dimension, we propose a spectral clustering dimensionality reduction strategy based on cosine similarity. Directly performing K-Means clustering on an HSI of size $C \times h \times w$ would face the curse of dimensionality. Therefore, as shown in Figure 2,we unfold the input features into a spectral matrix $M_{spe} \in \mathbb{R}^{C \times hw}$ and calculate the spectral similarity matrix $M_{sim} \in \mathbb{R}^{C \times C}$:

$$
M_{sim} = \text{Norm}(M_{spe}) \times \text{Norm}(M_{spe}^T). \tag{2}
$$

At this point, each row of $M_{sim}$ represents the cosine correlation of one band with other bands. This successfully reduces the dimensionality of the clustering features from $hw$ to $C$.

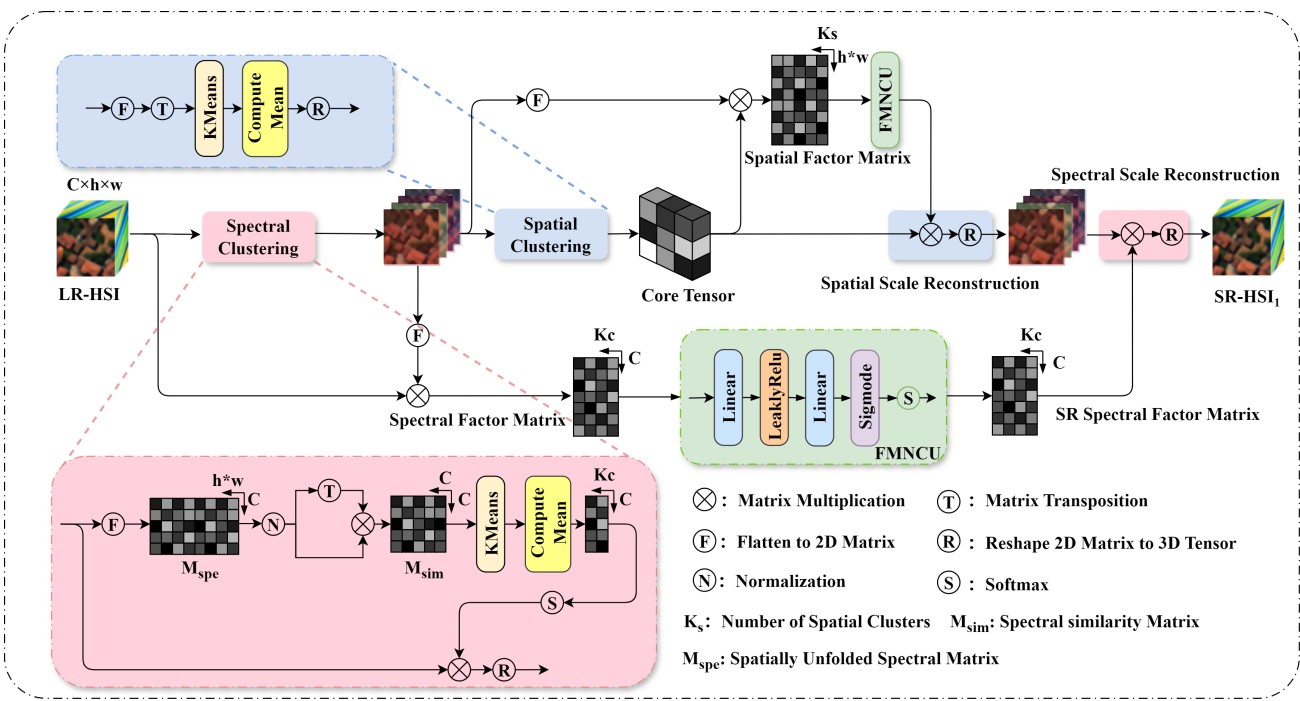

*Figure 2.* Detailed structure of the Spatial-spectral Dual-cluster Block (SSDCB).

In this work, K-Means is implemented as a hard clustering procedure (detailed steps are provided in the Appendix). Since the clustering is applied before any deep learning module in SSDCN, the discrete assignments do not block gradient flow to subsequent layers, and thus do not affect the model's training.

Performing K-Means clustering on $M_{sim}$ to divide it into $K_c$ clusters, and calculating the mean of each cluster yields the cluster centers. After Softmax transformation, multiplying it by the LR-HSI yields the $K_c$ spectral cluster center vectors of the LR-HSI. Because Softmax amplifies large values and suppresses small values, the Softmax transformation can enhance the representation ability of high-similarity spectra in the cluster center vectors and suppress a few distorted spectra. Then we fold it into a spatially dimensionality-reduced hyperspectral image of size $K_c \times h \times w$. This completes the dimensionality reduction in the spatial dimension.

The spectral cluster center feature $M_{spe\_red} \in \mathbb{R}^{K_c \times hw}$ and $M_{spe}$ are multiplied after normalization to obtain the spectral factor matrix $U_{spe} \in \mathbb{R}^{C \times K_c}$. $U_{spe}$ essentially consists of projection coefficients calculated from the low-resolution HSI. It is a low-resolution spectral factor matrix.

$$U_{spe} = \text{Norm}(M_{spe}) \times \text{Norm}(M_{spe\_red}^T). \quad (3)$$

**Factor Matrix Nonlinear Correction Unit (FMNCU):** $U_{spe}$ are essentially projection coefficients obtained through

linear projection from the low-resolution input. In order to enable them to have the ability to reconstruct high-resolution images, we designed the FMNCU module for nonlinear correction.

This unit comprises a sequence of Linear, LReLU, Linear, Sigmoid, and Softmax layers. It can learn a more complex nonlinear mapping function, meaning the model can autonomously decide which cluster features are more important, thereby making refined adjustments to the original projection coefficients. After the second Linear layer, the Sigmoid activation function is adopted, which can guarantee the non-negativity of the factor matrix. Moreover, the gentle saturation characteristic of Sigmoid at both ends can also play a role in noise suppression, pushing strong correlations towards 1 and suppressing irrelevant tiny projections to close to 0.

$U_I$ represents the factor matrix to be corrected, $H_{lin}$ represents the Linear layer, and $H_{FMNCU}$ represents the processing method of this module. Then the calculation process of FMNCU can be described as:

$$H_{FMNCU}(U_I) = \text{SoftMax}\big(\text{Sigmoid}\big( \\ H_{lin,2}(\text{LRelu}(H_{lin,1}(U_I))))\big). \quad (4)$$

**Spatial Clustering Dimensionality Reduction with Core Tensor and Spatial Factor Matrix:** We treat $M_{spe\_red}^T \in \mathbb{R}^{hw \times K_c}$ as spatial pixel vectors to be clustered. Since the dimension has been reduced to $K_c$, the clustering effi-

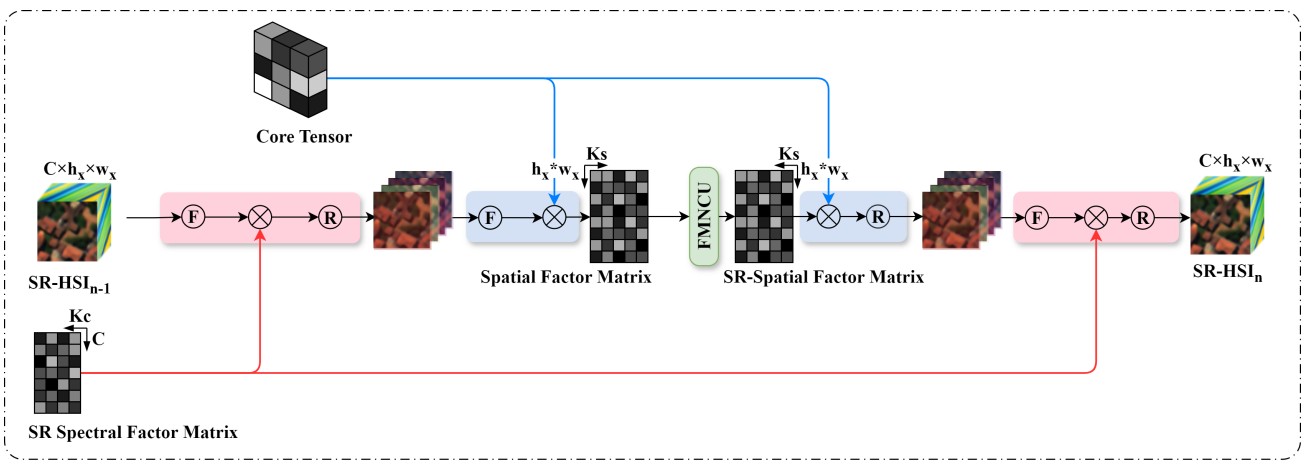

*Figure 3.* Detailed structure of the Feature Reuse Reconstruction Block (FRRB).

ciency here is extremely high. Through K-Means, they are clustered into $K_s$ cluster centers to obtain the core tensor $G \in \mathbb{R}^{K_s \times K_c}$,

$$G = \text{mean}(\text{Kmeans}(M^T_{spe\_red}, \dim = 0)). \quad (5)$$

$G$ is a highly compressed representation of the entire image, containing the most core, low-redundancy spatial-spectral joint features in the input LR-HSI. Similarly, the spatial factor matrix $U_{spa} \in \mathbb{R}^{hw \times K_s}$ is generated through the interaction between pixels and core features:

$$U_{spa} = M^T_{spe\_red} \times G^T. \quad (6)$$

$U_{spa}$ models the correlation of each pixel with global spatial information, effectively capturing long-range similar texture structures. $U_{spa}$ is also a low-resolution spatial factor matrix and needs to use FMNCU for nonlinear correction.

**Spatial-Spectral Reconstruction**: After successive spectral clustering dimensionality reduction and spatial clustering dimensionality reduction, we obtain $U_{spe}$ and $U_{spa}$. They are converted into super-resolution factor matrices $U_{SR\_Spe}$ and $U_{SR\_Spa}$ through FMNCU,

$$U_{SR\_Spe} = H_{FMNCU}(U_{spe}), \quad (7)$$
$$U_{SR\_Spa} = H_{FMNCU}(U_{spa}). \quad (8)$$

Now we need to reconstruct the super-resolution HSI based on $U_{SR\_Spe}$, $U_{SR\_Spa}$, and $G$. Matrix multiplication of $U_{SR\_Spa}$ and the transpose of $G$ yields the spatially reconstructed spatial matrix $M_{spa\_rec} \in \mathbb{R}^{K_c \times hw}$. Multiplying $M_{spa\_rec}$ and $U_{SR\_Spe}$ yields the spectrally reconstructed matrix $M_{rec} \in \mathbb{R}^{hw \times C}$. After folding and converting, the reconstructed hyperspectral image $I_{SR1} \in \mathbb{R}^{C \times h \times w}$ is obtained. $I_{SR\_x} \in \mathbb{R}^{C \times h_x \times w_x}$ represents the reconstructed features obtained after processing through x levels,where $h_x = 2^{x-1} \times h, w_x = 2^{x-1} \times w$,

$$M_{spa\_rec} = G^T \times U^T_{SR\_Spa}, \quad (9)$$
$$M_{rec} = M^T_{spa\_rec} \times U^T_{SR\_Spe}. \quad (10)$$

## 3.3. Feature Reuse Reconstruction Block (FRRB)

In subsequent scale levels ($\times 2$, $\times 4$, etc.), we no longer call the SSDCB module but instead use the designed FRRB module for hierarchical reconstruction. This module reuses $G$ and $U_{SR\_Spe}$ obtained from the $1\times$ scale level, as well as the $I_{SR\_n-1}$ input to this module, to recalculate the low-resolution spatial factor matrix under this scale, then corrects it through FMNCU, and reconstructs the super-resolution hyperspectral $I_{SR\_n}$.

As show in Figure 3, first, $I_{SR\_n-1}$ is unfolded along the spectral dimension into a spectral matrix $M_{spe\_n-1} \in \mathbb{R}^{C \times (h_x * h_x)}$. Multiplying the two matrices yields the spectrally dimensionality-reduced feature $M_{spe\_red\_n-1} \in \mathbb{R}^{K_c \times (h_x * h_x)}$. Similar to SSDCB, $G$ and $M_{spe\_red\_n-1}$ are directly multiplied to obtain the spatial factor matrix $U_{Spa\_n}$ of this scale level. FMNCU is used to nonlinearly correct it into the super-resolution spatial factor matrix $U_{SR\_Spa\_n}$. Subsequently, spatial and spectral reconstruction can be performed successively,

$$U_{Spa\_n} = M^T_{spe\_n-1} \times U_{SR\_Spe} \times G^T, \quad (11)$$
$$U_{SR\_Spa\_n} = H_{FMNCU}(U_{Spa\_n}). \quad (12)$$

## 3.4. Loss Function

To generate reconstructed HRHSI with enhanced spatial details and spectral fidelity, a joint loss function is defined to train SSDCN. This loss function includes four parts: reconstruction loss $\mathcal{L}_{rec}$, spectral fidelity loss $\mathcal{L}_{spe}$, gradient loss $\mathcal{L}_{gra}$, and multi-scale reconstruction loss $\mathcal{L}_{scale}$. The total loss is defined as:

$$\mathcal{L}_{total} = \mathcal{L}_{rec} + \alpha\mathcal{L}_{spe} + \beta\mathcal{L}_{gra} + \gamma\mathcal{L}_{scale}. \quad (13)$$

Where $\alpha = 0.003, \beta = 0.5, \gamma = 1$. $\mathcal{L}_{rec}, \mathcal{L}_{spe}, \mathcal{L}_{gra}$ are common loss terms in HSI-SISR. The multi-scale recon-

*Table 1.* Quantitative Comparison with SOTA Methods on the Pavia Center Dataset: Performance metrics (PSNR, SAM, ERGAS, SSIM, CC) at ×2, ×4, and ×8 scaling factors.

| Methods | 2× Scaling | | | | | 4× Scaling | | | | | 8× Scaling | | | | |
|---|---|---|---|---|---|---|---|---|---|---|---|---|---|---|---|
| | PSNR | SAM | ERGAS | SSIM | CC | PSNR | SAM | ERGAS | SSIM | CC | PSNR | SAM | ERGAS | SSIM | CC |
| Bicubic | 34.079 | 4.7695 | 4.6193 | 0.901 | 0.9446 | 28.8448 | 7.0668 | 8.2805 | 0.6987 | 0.8194 | 25.8229 | 9.593 | 11.7798 | 0.5369 | 0.6008 |
| DHP | 36.9808 | 4.3986 | 3.5156 | 0.9509 | 0.9605 | 30.685 | 6.4407 | 6.6482 | 0.8149 | 0.8831 | 26.6353 | 9.501 | 10.9616 | 0.6178 | 0.7097 |
| MCNet | 36.5016 | 4.2348 | 3.5927 | 0.9454 | 0.9619 | 30.8185 | 6.4178 | 6.6225 | 0.7946 | 0.8813 | 27.1181 | 8.8605 | 10.0555 | 0.6078 | 0.7246 |
| ERCSR | 37.1148 | 4.1091 | 3.3461 | 0.9516 | 0.9661 | 30.9526 | 6.4497 | 6.5168 | 0.8008 | 0.8836 | 26.7494 | 8.9714 | 10.5264 | 0.5846 | 0.6979 |
| MSDFormer | 37.2392 | 4.208 | 3.3908 | 0.9541 | 0.9633 | 30.8406 | 6.4899 | 6.5763 | 0.8046 | 0.8827 | 26.9699 | 8.806 | 10.2657 | 0.6097 | 0.7145 |
| ESSA | 37.5543 | 4.1269 | 3.3007 | 0.9555 | 0.9646 | 31.4355 | 6.0923 | 6.1593 | 0.8307 | 0.8958 | 27.0852 | 8.5905 | 10.0458 | 0.6278 | 0.7269 |
| CST | 37.1358 | 4.1769 | 3.4625 | 0.9538 | 0.9624 | 31.3724 | 6.1833 | 6.2520 | 0.8266 | 0.8944 | 27.2145 | 8.5733 | 9.9359 | 0.6188 | 0.7309 |
| SCPSN | _37.7238_ | _3.9015_ | _3.1983_ | _0.9578_ | _0.9679_ | _31.5756_ | _5.6875_ | _6.0688_ | _0.8339_ | _0.8989_ | _27.328_ | 8.2716 | 9.8175 | _0.6359_ | _0.7396_ |
| Res[3] | 36.9248 | 3.9264 | 3.4089 | 0.951 | 0.9652 | 31.1522 | 5.9333 | 6.3818 | 0.8073 | 0.8885 | 27.2694 | 8.4336 | _9.7992_ | 0.6175 | 0.7242 |
| SSRMamba | 37.2934 | 4.0413 | 3.3430 | 0.9539 | 0.9658 | 31.2248 | 5.9479 | 6.3251 | 0.8195 | 0.8919 | 27.2760 | _8.1738_ | 9.8998 | 0.6329 | 0.7369 |
| LDERT | 37.3805 | 3.9371 | 3.2906 | 0.9546 | 0.9665 | 31.1859 | 5.9000 | 6.3179 | 0.8158 | 0.8909 | 27.3019 | 8.4721 | 9.9763 | 0.6310 | 0.7381 |
| Ours | **38.1184** | **3.7004** | **3.051** | **0.9614** | **0.9705** | **31.7262** | **5.5207** | **5.9788** | **0.8394** | **0.9022** | **27.3647** | 7.9993 | **9.7889** | **0.6383** | **0.7417** |

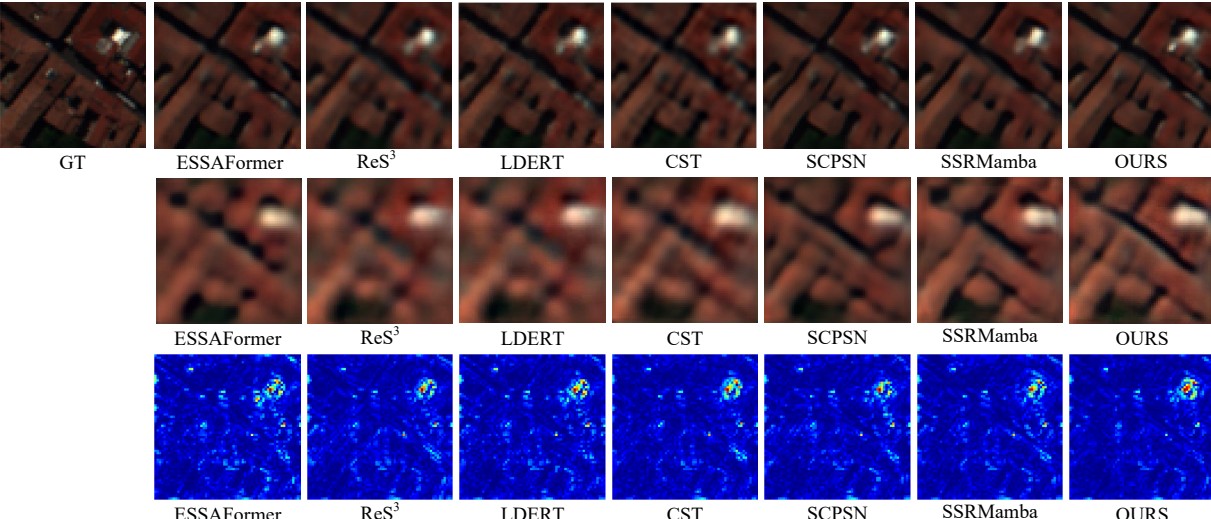

| | | | | | | |
|---|---|---|---|---|---|---|
| GT | ESSAFormer | ReS[3] | LDERT | CST | SCPSN | SSRMamba | OURS |

ESSAFormer  ReS[3]  LDERT  CST  SCPSN  SSRMamba  OURS

ESSAFormer  ReS[3]  LDERT  CST  SCPSN  SSRMamba  OURS

*Figure 4.* Visual comparison on the Pavia Center dataset. The first and second rows show the reconstruction results for 4× and 8× scaling factors, respectively. The third row displays the error difference maps (residuals) for the 4× SR task.

struction loss is defined as:

$$\mathcal{L}_{scale} = \sum_{x=1}^{\log_2 s} \mathcal{L}_{rec}^x. \qquad (14)$$

$s$ (2, 4, and 8) represents the model's scaling factors. $\mathcal{L}_{rec}^x$ represents the reconstruction loss at the ×$x$ scale level, specifically the L1 loss between the reconstruction result $I_{SR\_x}$ and $I_{GT\_bicubic(x)}$.

## 4. Experiments

### 4.1. Datasets and Experimental Settings

To demonstrate the performance of the proposed method, we conduct experiments on three commonly used hyperspectral datasets, including Pavia Center (Plaza et al., 2009), Botswana (Ungar, 2002), and Washington DC Mall (Aldrich

et al., 1996). All datasets are generated based on the Wald protocol. The ground-truth images are first blurred using an $8 \times 8$ Gaussian filter and then downsampled to $1/2$, $1/4$, and $1/8$ of the original size to obtain the low-resolution hyperspectral images (LR-HSI). For the three datasets, we crop them into 187, 92, and 80 non-overlapping cubes as ground truth (GT), and divide them into training and test sets with a ratio of 3:1.

For fair comparison, all deep learning-based methods are retrained on Ubuntu 20.04 using Python 3.9.13 and PyTorch 1.13.1 with an NVIDIA GeForce RTX A6000 GPU. The proposed SSDCN is trained using the Adam optimizer for 5000 epochs. The initial learning rate is set to $2 \times 10^{-4}$ with a batch size of 2. After training, the learning rate is reduced to $1 \times 10^{-5}$ with a batch size of 30 for an additional 100 epochs.

*Table 2.* Quantitative Comparison with SOTA Methods on the Botswana Dataset: Performance metrics at $\times 2$, $\times 4$, and $\times 8$ scaling factors.

| Methods | 2× Scaling | | | | | 4× Scaling | | | | | 8× Scaling | | | | |
|---|---|---|---|---|---|---|---|---|---|---|---|---|---|---|---|
| | PSNR | SAM | ERGAS | SSIM | CC | PSNR | SAM | ERGAS | SSIM | CC | PSNR | SAM | ERGAS | SSIM | CC |
| Bicubic | 40.0174 | 1.8687 | 2.1869 | 0.9005 | 0.9075 | 35.8657 | 2.8237 | 3.6900 | 0.7846 | 0.7496 | 33.6759 | 3.5515 | 4.8633 | 0.7193 | 0.5525 |
| DHP | 41.7317 | 1.7135 | 1.8493 | 0.9333 | 0.9339 | 36.5655 | 2.7133 | 3.5472 | 0.8196 | 0.8033 | 34.1217 | 3.4746 | 4.8313 | 0.7240 | 0.6426 |
| MCNet | 41.8825 | 1.5840 | 1.7622 | 0.9305 | 0.9364 | 36.4421 | 2.5950 | 3.5115 | 0.8211 | 0.7992 | 33.8576 | 3.4378 | 4.8036 | 0.7291 | 0.6153 |
| ERCSR | 41.1373 | 1.6867 | 1.9199 | 0.9300 | 0.9255 | 36.8997 | 2.6505 | 3.2308 | 0.8111 | 0.8142 | 33.8045 | 3.4962 | 4.8226 | 0.7148 | 0.5772 |
| MSDFormer | 41.7145 | 1.6412 | 1.8582 | 0.9348 | 0.9368 | 36.5454 | 2.6720 | 3.3940 | 0.8206 | 0.7917 | 34.1404 | 3.4251 | 4.6757 | 0.7383 | 0.6510 |
| ESSAFormer | 41.9300 | 1.6571 | 1.7788 | 0.9353 | 0.9374 | 37.5031 | 2.4944 | 3.0039 | 0.8281 | 0.8346 | 34.2975 | 3.4387 | 4.5252 | 0.7389 | 0.6392 |
| CST | 42.2682 | 1.5864 | 1.6754 | 0.9369 | 0.9406 | 37.6646 | 2.4000 | 2.9327 | 0.8282 | 0.8414 | 34.8612 | 3.2843 | 4.3313 | 0.7285 | 0.6876 |
| SCPSN | 42.5988 | 1.4990 | 1.5826 | 0.9371 | 0.9444 | 37.7980 | 2.3189 | 2.8759 | 0.8306 | 0.8449 | 35.1350 | 3.1120 | 4.1310 | 0.7421 | 0.7073 |
| Res[3] | 42.2351 | 1.5123 | 1.6358 | 0.9319 | 0.9404 | 37.0715 | 2.4588 | 3.1631 | 0.8272 | 0.8096 | 34.5852 | 3.2060 | 4.3538 | 0.7404 | 0.6489 |
| SSRMamba | 42.0777 | 1.5633 | 1.6746 | 0.9292 | 0.9387 | 36.9736 | 2.5074 | 3.1688 | 0.8212 | 0.8117 | 34.5399 | 3.2332 | 4.3659 | 0.7405 | 0.6506 |
| LDERT | 42.6514 | 1.4826 | 1.5742 | 0.9372 | 0.9454 | 37.6526 | 2.3318 | 2.9364 | 0.8291 | 0.8422 | 35.1646 | 3.0705 | 4.1282 | 0.7419 | 0.7076 |
| Ours | **42.8709** | **1.4524** | **1.5254** | **0.9397** | **0.9472** | **37.8910** | **2.2809** | **2.8459** | **0.8333** | **0.8477** | **35.1855** | **3.0556** | **4.0828** | **0.7429** | **0.7089** |

*Table 3.* Quantitative Comparison with SOTA Methods on the Washington DC Mall Dataset: Performance metrics at $\times 2$, $\times 4$, and $\times 8$ scaling factors.

| Methods | 2× Scaling | | | | | 4× Scaling | | | | | 8× Scaling | | | | |
|---|---|---|---|---|---|---|---|---|---|---|---|---|---|---|---|
| | PSNR | SAM | ERGAS | SSIM | CC | PSNR | SAM | ERGAS | SSIM | CC | PSNR | SAM | ERGAS | SSIM | CC |
| Bicubic | 49.1959 | 3.8568 | 4.6439 | 0.9527 | 0.9427 | 44.2918 | 7.0875 | 8.2807 | 0.8686 | 0.8025 | 41.6946 | 9.8181 | 12.6389 | 0.8114 | 0.5581 |
| DHP | 48.6770 | 4.1999 | 5.4657 | 0.9608 | 0.9337 | 44.2277 | 6.0418 | 11.9932 | 0.9087 | 0.8009 | 41.6603 | 8.9784 | 12.9644 | 0.8343 | 0.6283 |
| MCNet | 49.3561 | 3.2657 | 6.7956 | 0.9631 | 0.9336 | 44.8932 | 6.0256 | 9.4150 | 0.9017 | 0.8342 | 41.8294 | 8.7748 | 13.0103 | 0.8364 | 0.6504 |
| ERCSR | 50.6414 | 3.0378 | 4.5261 | 0.9702 | 0.9517 | 45.5203 | 5.8795 | 7.4903 | 0.9053 | 0.8515 | 41.7427 | 8.8516 | 14.5249 | 0.8339 | 0.6419 |
| MSDFormer | 50.3477 | 3.2030 | 4.5819 | 0.9709 | 0.9499 | 45.3878 | 6.2597 | 7.2885 | 0.8958 | 0.8498 | 42.5329 | 9.0397 | 10.1801 | 0.8361 | 0.6821 |
| ESSAFormer | 51.2354 | 2.7663 | 3.9563 | 0.9730 | 0.9607 | 45.9792 | 5.2684 | 6.9037 | 0.9161 | 0.8689 | 42.1159 | 8.7289 | 10.8149 | 0.8323 | 0.6568 |
| CST | 51.5148 | 2.8717 | 4.0666 | 0.9723 | 0.9624 | 46.2649 | 5.8130 | 6.5021 | 0.9179 | 0.8775 | 42.5333 | 8.8424 | 10.1331 | 0.8326 | 0.6822 |
| SCPSN | 51.4377 | 2.6749 | 3.6909 | 0.9778 | 0.9627 | 46.4058 | 5.0035 | 6.4372 | 0.9194 | 0.8799 | 42.6109 | 8.2052 | 10.0698 | 0.8366 | 0.6939 |
| Res[3] | 51.2863 | 2.8648 | 3.7171 | 0.9697 | 0.9625 | 45.9711 | 5.5918 | 6.7373 | 0.9067 | 0.8668 | 42.5629 | 8.5790 | 10.0898 | 0.8351 | 0.6830 |
| SSRMamba | 50.9183 | 3.0802 | 3.8015 | 0.9702 | 0.9592 | 45.7699 | 5.7821 | 6.9338 | 0.9045 | 0.8618 | 42.5897 | 8.7965 | 10.1190 | 0.8360 | 0.6874 |
| LDERT | 51.0725 | 2.7879 | 4.1628 | 0.9718 | 0.9578 | 45.7322 | 5.5846 | 7.0234 | 0.9071 | 0.8597 | 42.6091 | 8.3799 | 10.0200 | 0.8378 | 0.6854 |
| Ours | **52.7285** | **2.3425** | **3.4666** | **0.9809** | **0.9707** | **46.4789** | **4.9508** | **6.3740** | **0.9198** | **0.8818** | **42.6903** | **8.1974** | **9.9744** | **0.8400** | **0.6941** |

## 4.2. Comparison with SOTA Methods

To evaluate the effectiveness of SSDCN, we compare it with ten state-of-the-art methods (including Transformers like CST (Chen et al., 2024) , ESSAFormer (Zhang et al., 2023) , LDERT (Li et al., 2025b) , MSDFormer (Chen et al., 2023) , SCPSN (Yang et al., 2024) and recent models like SSRMamba (Li et al., 2025a) , ReS[3] (Hou et al., 2024) ,ERCSR (Li et al., 2021) , MCNet (Li et al., 2020) , DHP (Sidorov & Hardeberg, 2019)) using five standard metrics: PSNR, SSIM, SAM, ERGAS, and CC. Tables 1, 2, and 3 summarize the quantitative results across three datasets.

As shown in Tables 1, 2, and 3, SSDCN consistently achieves superior performance over the competing methods across the three benchmark datasets. On the **Pavia Center** and **Botswana** datasets, our method ranks first across all scales (2×, 4×, and 8×) and all evaluation metrics, demonstrating its strong capability in both spatial reconstruction and spectral fidelity preservation. On the **Washington DC Mall** dataset, SSDCN also obtains the best results on all metrics at the 2×, 4×, and 8× scales. These consistent improvements indicate that SSDCN remains robust under

*Table 4.* Comparison with variants removing both SSDCB and FRRB and variants removing only the FRRB.

| Method | FLOPs (G) | PSNR | SAM | ERGAS |
|---|---|---|---|---|
| w/o SSDCB & FRRB | **3.0511** | 31.5732 | 5.6019 | 6.0752 |
| w/o FRRB module | 3.0554 | 31.6276 | 5.6010 | 6.0429 |
| SSDCN (Complete) | 3.0680 | **31.7262** | **5.5207** | **5.9788** |

different scenes and magnification factors, especially in challenging large-scale reconstruction settings.

Figure 4 presents a visual comparison on the Pavia Center dataset. At both and scales, our SSDCN recovers significantly sharper details and edges compared to the blurred results from competitors like ESSAFormer and CST. The error maps in the third row further confirm our method's superiority, exhibiting the lowest error magnitude and demonstrating high spatial-spectral fidelity.

## 4.3. Ablation Experiments

To verify the effectiveness and rationality of each component in SSDCN, we perform extensive ablation studies on the

*Table 5.* Comparison with variants removing Spectral Clustering (Only Space Dim. Red.), Spatial Clustering (Only Channel Dim. Red.), or both (Neither Reduced).

| Variant | FLOPs (G) | PSNR | SAM | ERGAS |
|---|---|---|---|---|
| Only Channel Dim. Red. | 3.7662 | 31.6035 | 5.5518 | 6.0455 |
| Only Space Dim. Red. | 3.1422 | 31.6737 | 5.5424 | 6.0308 |
| Neither Reduced | 3.8475 | 31.6856 | 5.5430 | 5.9950 |
| SSDCN (Ours) | **3.0680** | **31.7262** | **5.5207** | **5.9788** |

*Table 6.* Comparison with variants reusing only the Core Tensor, the Spectral Factor Matrix, or no reuse (standard SSDCB).

| Variant | FLOPs (G) | PSNR | SAM | ERGAS |
|---|---|---|---|---|
| SSDCB Replace FRRB | 3.0963 | 31.5871 | 5.5865 | 6.0593 |
| Only Reuse Core Tensor | 3.0888 | 31.3866 | 5.6993 | 6.2142 |
| Only Reuse Spe. Factor | 3.0754 | 31.6084 | 5.5867 | 6.0578 |
| SSDCN (Complete) | **3.0680** | **31.7262** | **5.5207** | **5.9788** |

Pavia Center dataset under $4\times$ super-resolution.

**Model Component Ablation**. As shown in Table 4, removing both SSDCB and FRRB modules leads to a sharp performance drop, with PSNR decreasing to only 31.57 dB. The variant without FRRB shows clear improvement over the baseline, validating the effectiveness of SSDCB. However, the full model still significantly outperforms both variants, demonstrating the importance of the FRRB module.

**Internal Ablation of SSDCB Module**. To prove the effectiveness of the internal design of the SSDCB module, we performed removal tests on the spatial-spectral clustering module. As shown in Table 5, removing dimensionality reduction modules leads to a surge in model parameters and calculation volume—most notably when the spatial clustering module is removed—and performance metrics decline. This is because redundant spectra and some spatial noise pixels interfere with the reconstruction effect.

**Internal Ablation of FRRB Module**. As shown in Table 6, the complete reuse strategy achieves optimal performance while maintaining the lowest computational cost. Notably, the variant reusing only the core tensor exhibits the most significant performance decline. This is because the core tensor, serving as a compressed representation of global features, implicitly encodes specific central spectral information; forcibly combining it with a spectral factor matrix derived from re-clustering leads to a mismatch in feature space distribution, thereby inducing training instability and performance deterioration.

**Loss Function Ablation**. As shown in Table 7, removing any loss component degrades performance, especially the gradient loss and multi-scale hierarchical reconstruction loss, which cause a PSNR drop of nearly 0.2 dB.

*Table 7.* Ablation Study on Loss Functions

| $\mathcal{L}_{rec}$ | $\mathcal{L}_{spe}$ | $\mathcal{L}_{gra}$ | $\mathcal{L}_{scale}$ | PSNR | SAM |
|---|---|---|---|---|---|
| ✓ | | | | 31.4925 | 5.9180 |
| ✓ | | ✓ | ✓ | 31.6341 | 5.8673 |
| ✓ | ✓ | | ✓ | 31.5534 | 5.6536 |
| ✓ | ✓ | ✓ | | 31.5262 | 5.7129 |
| ✓ | ✓ | ✓ | ✓ | **31.7262** | **5.5207** |

*Table 8.* Efficiency vs. Performance Comparison

| Method | Params (M) | FLOPs (G) | PSNR (dB) |
|---|---|---|---|
| DHP | 8.58 | 4.65 | 30.6850 |
| MCNet | 2.17 | 230.51 | 30.8185 |
| ERCSR | 1.59 | 229.30 | 30.9526 |
| MSDFormer | 32.99 | 24.07 | 30.8406 |
| ESSAFormer | 11.52 | 50.19 | 31.4355 |
| ReS$^3$ | 4.74 | 193.09 | 31.1522 |
| SCPSN | 3.81 | 6.37 | 31.5756 |
| CST | 10.99 | 4.97 | 31.3724 |
| LDERT | 92.78 | 15.62 | 31.1859 |
| SSRMamba | 8.31 | **1.75** | 31.2248 |
| SSDCN (Ours) | **1.31** | 3.07 | **31.7262** |

*Table 9.* End-to-End Runtime Comparison with Transformer-based Methods on the Pavia Center Dataset at $\times 4$ Scale.

| Method | Time (s) | Peak Memory (GB) |
|---|---|---|
| ESSAFormer | 0.5702 | 0.3228 |
| CST | 0.6914 | 0.0653 |
| SCPSN | 0.6242 | 0.0378 |
| LDERT | 0.6782 | 0.7968 |
| Ours | **0.5139** | **0.0293** |

### 4.4. Model Complexity

To evaluate the efficiency-performance trade-off, we compare the parameter count, computational cost, and reconstruction quality of SSDCN with other SOTA methods on the Pavia Center dataset ($\times 4$ scale). As shown in Table 8, SSDCN requires only 1.31M parameters and 3.07G FLOPs, which is significantly lower than Transformer-based models. Although our computational cost is slightly higher than SSRMamba, our method achieves a 0.5 dB higher PSNR, demonstrating an excellent balance between efficiency and performance.

To further evaluate the practical efficiency of SSDCN, Table 9 reports the end-to-end inference time and peak GPU memory consumption. Compared with representative Transformer-based methods, SSDCN achieves the best efficiency in both runtime and memory usage, requiring only 0.5139 seconds and 0.0293 GB peak memory. This demonstrates that the proposed method provides a lightweight and efficient inference process while maintaining strong reconstruction performance.

# 5. Conclusion

To address the challenge of balancing global modeling capability and computational efficiency in HSI-SISR, this paper proposes an efficient network based on spatial-spectral dual clustering tensor decomposition, named SSDCN.Its core component, SSDCB, innovatively employs a clustering mechanism to substitute traditional point-to-point self-attention calculations. This design reduces the complexity of global modeling from quadratic to linear, effectively capturing spatial non-local self-similarity and spectral low-rank properties while maintaining extremely low computational overhead. The model adopts a pyramid progressive architecture and utilizes a FRRB to share the core tensor and spectral factor matrix across different scale levels, dynamically updating only the spatial dimension.This significantly reduces computational redundancy in deep networks. Experimental results demonstrate that SSDCN achieves SOTA performance in terms of reconstruction accuracy and spectral fidelity on several mainstream datasets, such as Pavia Center. Furthermore, with only 1.31M parameters and 3.07G FLOPs, its computational cost is significantly lower than existing Transformer-based models.

# Acknowledgements

This work was supported in part by the Natural Science Foundation of Tianjin under Grant 24JCZDJC00130 and Grant 25JCZDJC00540, in part by the Natural Science Foundation of Hebei under Grant F2025110006 and Grant F2025110010, and in part by the Cangzhou Institute of Tiangong University under Grant TGCYY-Z-0303.

# Impact Statement

This paper presents work whose goal is to advance the field of Machine Learning. There are many potential societal consequences of our work, none of which we feel must be specifically highlighted here.

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

## A. Appendix Outline

In this supplementary material, we provide more details of our SSDCN as follows:

- **Sec. B** analyzes the spectral low-rank properties and non-local self-similarity of hyperspectral images (HSI) to validate the motivation of our method.

- **Sec. C** demonstrates the visual evolution process of the pyramid progressive hierarchical reconstruction architecture.

- **Sec. D** details the implementation of the K-Means clustering algorithm used in SSDCN.

- **Sec. E** details the mathematical formulations of the conventional loss functions used in our training phase.

- **Sec. F** introduces the specific details of the datasets (Pavia Center, Botswana, and Washington DC Mall) and the experimental settings.

- **Sec. G** presents additional visual comparison results and difference maps.

- **Sec. H** reports supplementary ablation experiments, including discussions on dimensionality reduction strategies, model architecture, and hyperparameter selection ($K_c$, $K_s$).

- **Sec. I** presents the SR-HSI classification experiments to verify the effectiveness of the proposed method on downstream tasks.

## B. Spectral Low-Rank Properties and Non-Local Self-Similarity of Hyperspectral Images

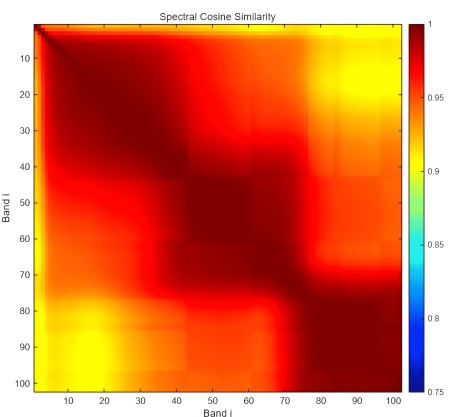
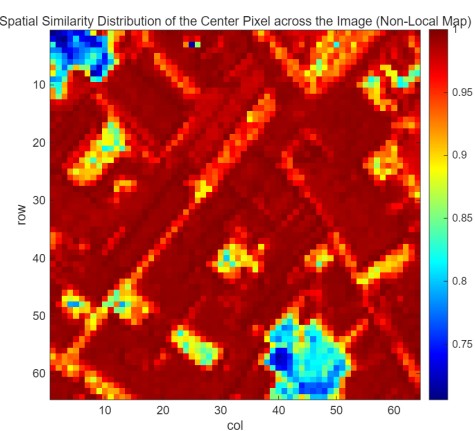

*Figure 5.* Visualization of Correlation Properties in Hyperspectral Images. Left: Spectral Cosine Similarity Heatmap indicating strong inter-band correlations. Right: Spatial Similarity Distribution of the Center Pixel (Non-Local Map) revealing global self-similarity.

As shown in Figure 5 (Left), we calculated the cosine similarity between different bands of a hyperspectral image (HSI) and visualized it. The large areas of high intensity (red/yellow) in the heatmap indicate that there is an extremely strong correlation among the hundreds of bands in the HSI. Not only are adjacent bands highly similar (the correlation of adjacent bands approaches 1, represented by the dark red diagonal line in Figure 5), but even bands that are far apart often exhibit significant correlation characteristics. This high degree of redundancy intuitively confirms that hyperspectral data essentially resides in an extremely low-dimensional subspace.

To verify the distribution characteristics of spatial textures in HSI, we randomly selected a pixel in the center of the image as a query point and calculated its similarity with all other pixels in the entire image. The results are shown in Figure 5 (Right). It can be clearly observed that regions highly similar to the central pixel (highlighted parts) are not confined to its immediate neighborhood but are discretely distributed across the global range of the image. This phenomenon reveals that HSI possesses significant non-local self-similarity. It strongly highlights the limitations of Window-based or local convolutional neural network (CNN) methods—namely, aggregating features only within a local receptive field inevitably leads to the loss of long-range key texture details.

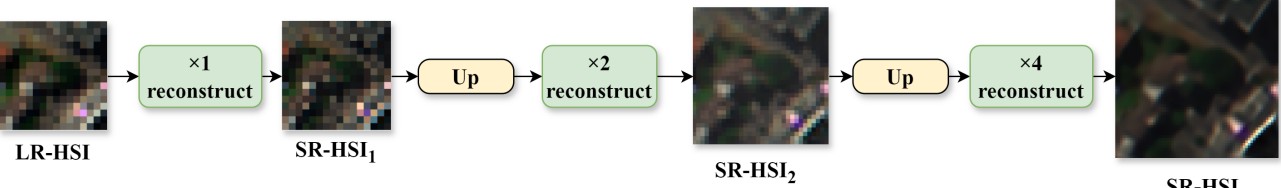

*Figure 6.* Illustration of the Pyramid Progressive Hierarchical Reconstruction Process. This strategy allows for coarse-to-fine detail recovery.

## C. Demonstration of Pyramid Progressive Hierarchical Reconstruction

Figure 6 intuitively illustrates the super-resolution reconstruction process of the SSDCN model under the pyramid progressive architecture. Unlike traditional direct End-to-End magnification, SSDCN adopts a "coarse-to-fine" stepwise generation strategy. As seen in Figure 6, after the first level of reconstruction, the high-frequency features of the HSI image are enriched. Simultaneously, due to the correction by the spectral factor matrix, the spectral information becomes more realistic. As the levels progress, the edges of the image gradually become sharper, and originally blurred texture details (such as building contours and road boundaries) are progressively resolved. This level-by-level supervision mechanism (combined with the pyramid hierarchical reconstruction joint loss) effectively suppresses the error accumulation problem common in deep networks, ensuring that the final output SR-HSI maintains both physical structural authenticity and high-fidelity spectral characteristics.

## D. Implementation Details of K-Means Clustering

In this work, we adopt the classical K-Means algorithm as a hard clustering strategy, where sample assignment is performed based on the Euclidean distance. Given a set of input feature vectors $\mathcal{X} = \{x_i\}_{i=1}^{N}$ and the number of clusters $K$, K-Means alternates between sample assignment and cluster center updating. Specifically, each sample is assigned to its nearest cluster center:

$$z_i = \arg \min_{k \in \{1,...,K\}} \|x_i - \mu_k\|_2^2, \tag{15}$$

where $\mu_k$ denotes the center of the $k$-th cluster and $z_i$ is the hard cluster assignment of $x_i$. Then, each cluster center is updated by averaging all samples assigned to it:

$$\mu_k = \frac{1}{|\mathcal{C}_k|} \sum_{x_i \in \mathcal{C}_k} x_i, \tag{16}$$

where $\mathcal{C}_k$ denotes the set of samples belonging to the $k$-th cluster.

In SSDCN, K-Means is used in both spectral clustering and spatial clustering. For spectral clustering, the input samples are the band-wise similarity vectors derived from the spectral similarity matrix. For spatial clustering, the input samples are the pixel features after spectral dimensionality reduction. It is worth noting that although the spectral similarity matrix is constructed using cosine similarity, the clustering process itself still follows the standard Euclidean-distance-based K-Means algorithm.

To reduce the additional computational overhead, the maximum number of K-Means iterations is set to 5 in our implementation. The time complexity of standard K-Means is $\mathcal{O}(TNKD)$, where $T$ is the number of iterations, $N$ is the number of samples, $K$ is the number of clusters, and $D$ is the feature dimension. In this work, both $T$ and $K$ are predefined hyperparameters, and the number of clusters $K$ is much smaller than the number of samples $N$. Therefore, the additional computational cost introduced by K-Means is controllable. In particular, for spatial clustering, the pixel features have already been spectrally reduced, which further decreases the feature dimension $D$ and reduces the clustering cost.

## E. Conventional Loss Functions for HSI-SISR

Numerous studies have shown that $L_1$ and $L_2$ losses are effective in SR tasks. However, it has been proven that $L_1$ loss performs better as a reconstruction loss compared to $L_2$ loss. Therefore, we use the $L_1$ loss to calculate the pixel-level difference between the reconstructed HR-HSI, $I_{SR}$, and the corresponding ground truth (GT) image, $I_{GT}$, which can be

expressed as:

$$\mathcal{L}_{rec} = \frac{1}{M} \sum_{m=1}^{M} \|I_{GT}^m - I_{SR}^m\|_1. \tag{17}$$

where $M$ represents the number of images in a training batch, and $\| \cdot \|_1$ denotes the $L_1$ norm.

To ensure the spectral fidelity of the reconstructed HR-HSI, a spectral fidelity loss is defined to measure the spectral similarity between $I_{SR}$ and $I_{GT}$. The spectral fidelity loss is expressed as:

$$\mathcal{L}_{spe} = \frac{1}{M} \sum_{m=1}^{M} \frac{1}{\pi} \arccos \left( \frac{I_{GT}^m \cdot I_{SR}^m}{\|I_{GT}^m\|_2 \cdot \|I_{SR}^m\|_2} \right). \tag{18}$$

where $\arccos(\cdot)$ denotes the arccosine function, and $\| \cdot \|_2$ denotes the $L_2$ norm.

To ensure that the reconstructed HR-HSI possesses rich high-frequency features, a gradient loss is defined to constrain the consistency of spatial gradients between the reconstructed image and the real image. This loss reinforces the model's ability to recover edge and texture details by calculating the difference in gradient magnitude in the horizontal and vertical directions. Specifically, the Sobel operator is adopted as the gradient operator, with convolution kernels as follows:

$$\text{Horizontal Gradient Operator } G_x = \begin{bmatrix} -1 & 0 & 1 \\ -2 & 0 & 2 \\ -1 & 0 & 1 \end{bmatrix}, \quad \text{Vertical Gradient Operator } G_y = \begin{bmatrix} -1 & -2 & -1 \\ 0 & 0 & 0 \\ 1 & 2 & 1 \end{bmatrix}. \tag{19}$$

For an image $I$, its gradient magnitude map $G(I)$ can be calculated as:

$$G(I) = \sqrt{(I * G_x)^2 + (I * G_y)^2}, \tag{20}$$

where $*$ denotes the convolution operation. The gradient loss $\mathcal{L}_{gra}$ is defined as the $L_1$ norm difference between the gradient magnitude maps of the reconstructed image $I_{SR}$ and the real image $I_{GT}$:

$$\mathcal{L}_{gra} = \frac{1}{M} \sum_{m=1}^{M} \|G(I_{GT}^m) - G(I_{SR}^m)\|_1. \tag{21}$$

## F. Dataset Introduction

**Pavia Center Dataset:** The Pavia Center dataset was acquired by the ROSIS sensor during a flight campaign over Pavia, northern Italy. 13 noisy bands were discarded from the original HSI, resulting in an HSI with 102 bands (430 nm to 860 nm). Additionally, a rectangular area of $1096 \times 381$ pixels with no information in the center of the original HSI was discarded, yielding a "two-part" image of size $1096 \times 715 \times 102$ for the experiment. This was divided into 187 non-overlapping cubes of size $64 \times 64 \times 102$ to obtain the GT for the Pavia Center dataset. 138 cubes were randomly selected for training, and the remaining 49 cubes were used for testing.

**Botswana Dataset:** The Botswana scene was acquired by the Hyperion sensor on NASA's Earth Observing-1 (EO-1) satellite. The original Botswana HSI consists of 242 spectral bands ranging from 400 to 2500 nm, with a spectral resolution of 10 nm. The spatial size of the original image is $1496 \times 256$ pixels. We removed uncalibrated and noisy bands from the original image to obtain an HSI with 145 bands. This was further divided into 92 non-overlapping cubes of size $64 \times 64 \times 145$ to obtain the GT for the Botswana dataset. 69 cubes were randomly selected for training, and the remaining 23 cubes were used for testing.

**Washington DC Mall Dataset:** The Washington DC Mall dataset was acquired by the Hyperspectral Digital Imagery Collection Experiment (HYDICE) sensor. The original HSI consists of 191 spectral bands ranging from 400 to 2500 nm. The spatial size of the original Washington DC Mall image is $1208 \times 307$ pixels. This dataset was further divided into 80 partially overlapping cubes of size $64 \times 64 \times 191$ ($1208 \times 307$ can be cut into at most 72 non-overlapping cubes; the last 8 cubes have minor overlaps with others but were placed in the training set) to obtain the GT for the Washington DC Mall dataset. Among them, 60 cubes were randomly selected for training, and the remaining 20 cubes were used for testing.

To display the experimental results, three spectral bands were selected from the HSI to generate RGB images. For the HSI in the Botswana, Washington DC Mall, and Pavia Center datasets, we selected the 61st, 60th, and 60th spectral channels as the red channel; the 10th, 27th, and 10th spectral channels as the blue channel; and the 35th, 17th, and 30th spectral channels as the green channel, respectively.

## G. Additional Visual Comparisons

This chapter supplements the visual comparison with SOTA methods. Figure 8 presents the visual results for the Pavia center dataset (4× and 8× scales) for methods omitted from the main text, with residual maps shown in the bottom row. Our method recovers significantly more high-frequency details and achieves better visual fidelity. Notably, our residual maps contain fewer bright artifacts than other methods, reflecting the lowest error rates. Figures 9–11 further showcase the superior restoration capacity of our model on Pavia Center (2×), Botswana (4×), and Washington DC (4×). Additionally, Figures 7 and 12 illustrate the average spectral difference across datasets. The red line (our model) consistently remains at the lowest position in all plots, confirming that our model achieves the minimum spectral error between the SR-HSI and GT across every band.

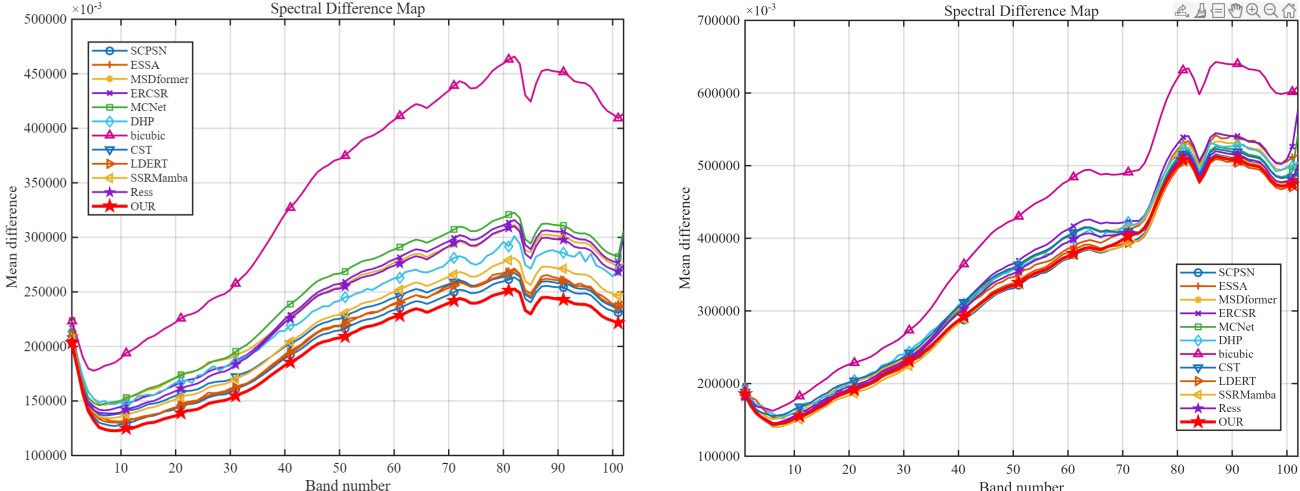

*Figure 7.* Mean Spectral Difference Maps on Pavia Center Dataset. Left: Results for ×4 Scale. Right: Results for ×8 Scale.

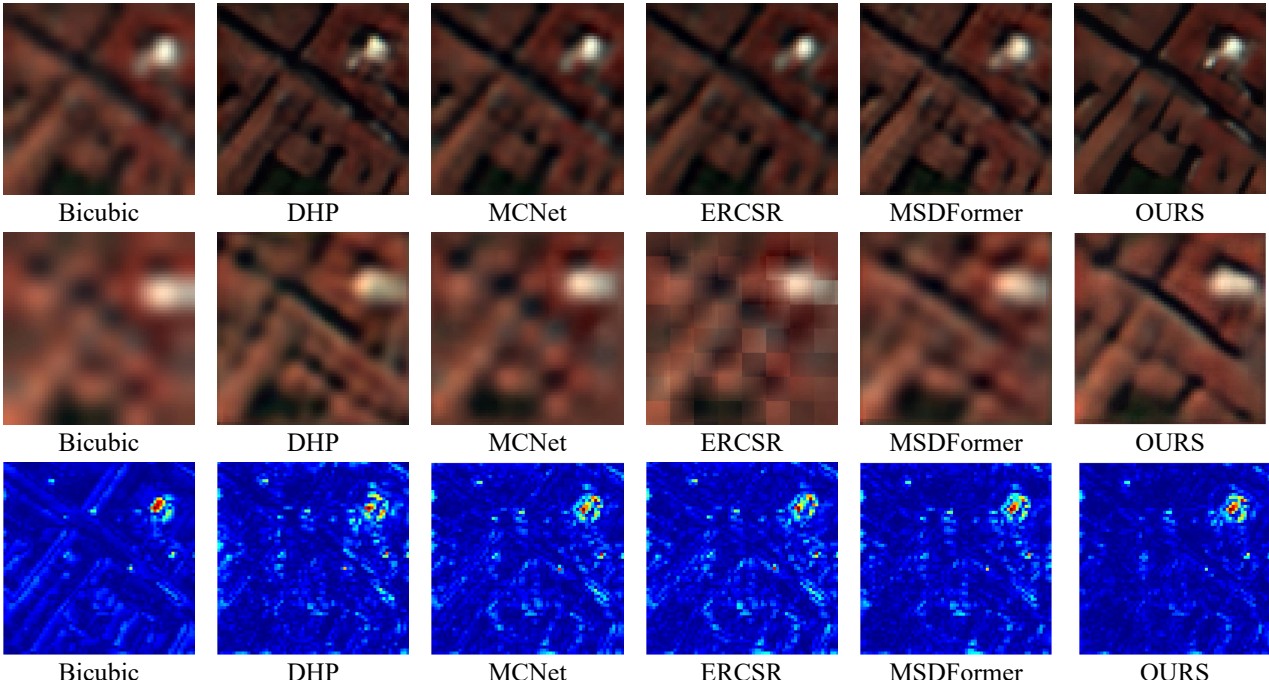

*Figure 8.* Supplementary Visual Comparisons. Top: Pavia Center ×4 and ×8 results. Bottom: Residual maps for Pavia Center ×4 scale (Models not shown in main text).

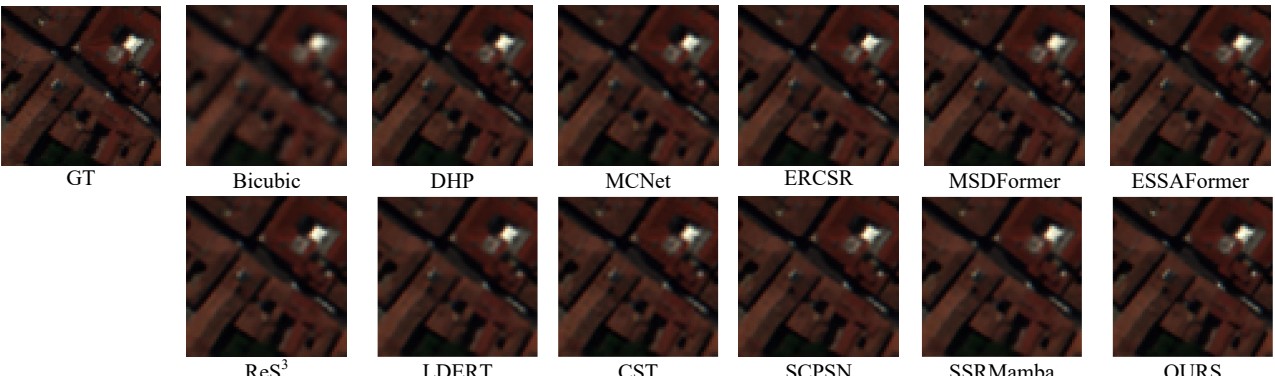

*Figure 9.* Visual Comparison of All Models on Pavia Center Dataset (×2 Scale).

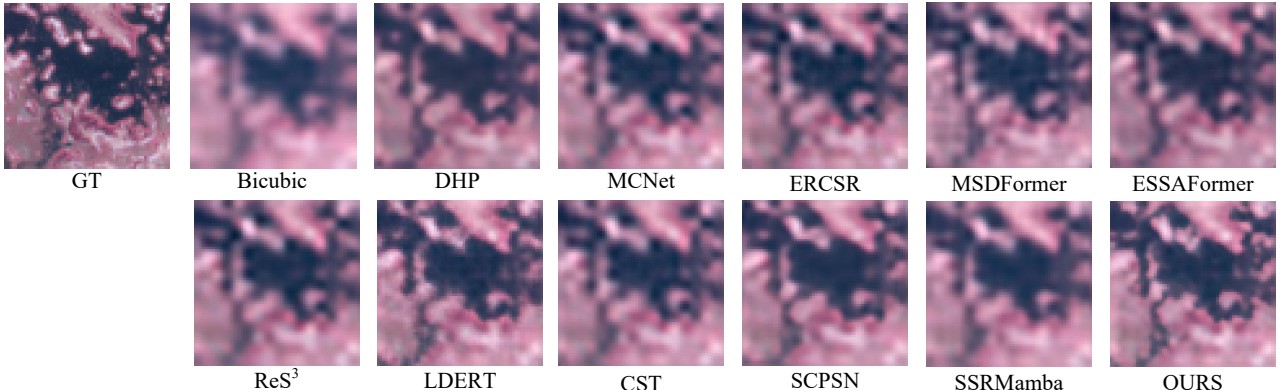

*Figure 10.* Visual Comparison of All Models on Botswana Dataset (×4 Scale).

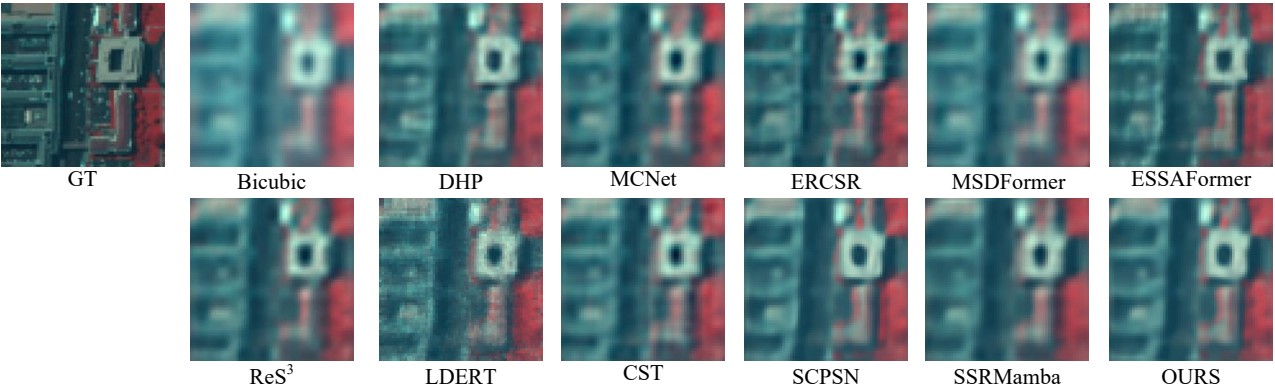

*Figure 11.* Visual Comparison of All Models on Washington DC Mall Dataset(×4 Scale).

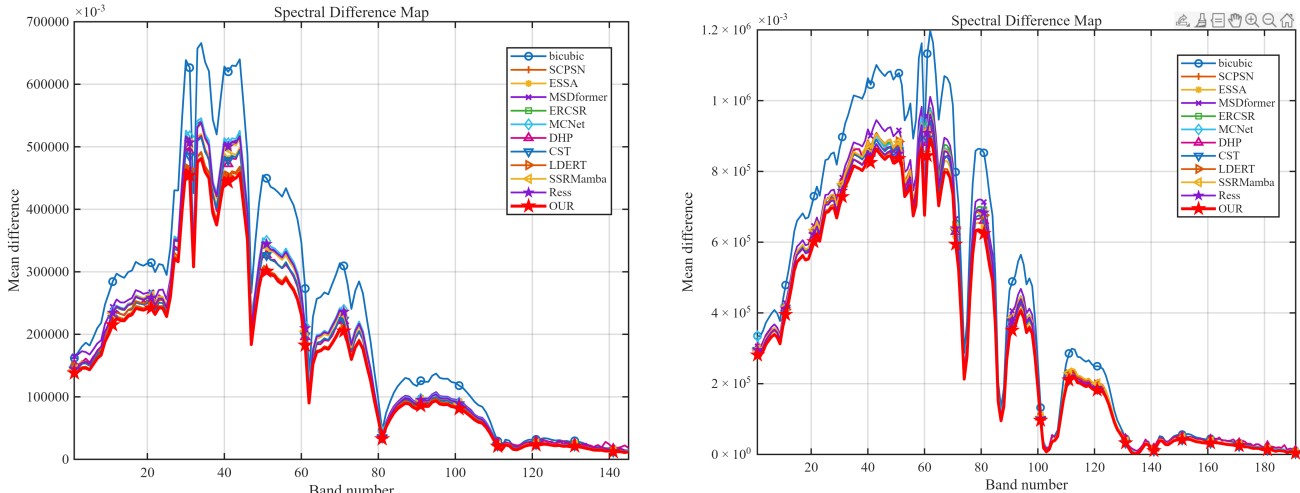

*Figure 12.* Mean Spectral Difference Maps (×4 Scale). Left: Botswana Dataset. Right: Washington DC Mall Dataset.

## H. Supplementary Ablation Experiments

**Ablation of Spatial-Spectral Dimensionality Reduction Methods:** Our model uses a K-Means clustering-based method to achieve spatial and spectral dimensionality reduction. The advantage of clustering is that it can utilize global similarity information, fully exploiting the spectral low-rank properties and spatial non-local similarity characteristics of hyperspectral images. To highlight the advantages of clustering-based dimensionality reduction, we designed four model variants, as shown in Table 10. "Random Selection" represents randomly selecting spectra and spatial pixels as the center spectra and center spatial pixels after dimensionality reduction. We used $1 \times 1$ convolution to map all channels to $K_c$ channels to replace spectral clustering, and adaptive global average pooling to replace spatial clustering. We combined these with the original method to obtain the model variants in the second to fourth rows. As can be seen from Table 10, whether using random selection or dimensionality reduction strategies based on convolution or pooling, the model performance declines. Specifically, using Adaptive Global Average Pooling to replace spatial clustering as the spatial dimensionality reduction method results in the most significant drop in metrics. This is because global average pooling only utilizes local information and cannot fully leverage the global non-local similarity characteristics associated with hyperspectral images.

*Table 10.* Ablation Study on Spatial-Spectral Dimensionality Reduction Strategies

| Method | FLOPs (G) | PSNR | SAM | ERGAS |
|---|---|---|---|---|
| Random Selection | **3.0539** | 31.5992 | 5.6113 | 6.0512 |
| $1 \times 1$ Conv (Spe) + Kmeans (Spa) | 3.0577 | 31.6280 | 5.5695 | 6.0342 |
| Kmeans (Spe) + AvgPooling (Spa) | 3.0643 | 31.4126 | 5.6967 | 6.1902 |
| $1 \times 1$ Conv (Spe) + AvgPooling (Spa) | 3.0540 | 31.6149 | 5.5947 | 6.0510 |
| **Kmeans Dual-Clustering (Ours)** | 3.0680 | **31.7262** | **5.5207** | **5.9788** |

**Ablation of Training Strategy:** Our model adopts a two-stage training strategy to improve optimization stability. As shown in Table 11, compared with StepLR and Cosine Annealing, the proposed strategy achieves the best overall performance in most metrics. This indicates that the two-stage training strategy provides a more stable optimization process and improves the reconstruction quality of SSDCN.

*Table 11.* Ablation Study on Training Strategies

| Training Strategy | PSNR | SSIM | SAM | ERGAS | RMSE | CC |
|---|---|---|---|---|---|---|
| StepLR | 31.6926 | 0.8380 | 5.5736 | 5.9832 | **0.0316** | 0.9021 |
| Cosine Annealing | 31.6831 | 0.8376 | 5.5623 | 5.9964 | 0.0317 | 0.9010 |
| Ours | **31.7262** | **0.8394** | **5.5207** | **5.9788** | **0.0316** | **0.9022** |

**Ablation of Model Progressive Pyramid Architecture:** The pyramid progressive scale level reconstruction architecture

differs from other HSI-SISR model architectures by adopting a progressive upsampling method for level-by-scale reconstruction. To prove the superiority of this architecture over other model architectures, we designed a model variant. As shown in Table 12, "Direct ×4" represents that relative to the full model, after the ×1 scale processing, it is directly magnified to the target scale for processing. To ensure fairness in the experiment, we did not remove the hierarchical reconstruction loss associated with this architecture. As seen in Table 12, although the parameter count of the variant model is reduced compared to the full model, the performance of the model drops significantly; the PSNR metric decreases by 0.26 dB, and the SAM metric rises to nearly 5.7. This demonstrates the superiority of the progressive upsampling model architecture proposed in this paper.

*Table 12.* Ablation Study on Model Progressive Pyramid Architecture

| Method | FLOPs (G) | PSNR | SAM | ERGAS |
|---|---|---|---|---|
| Direct ×4 | **2.7380** | 31.4619 | 5.6915 | 6.1493 |
| **Progressive Pyramid Architecture (Ours)** | 3.0680 | **31.7262** | **5.5207** | **5.9788** |

**Discussion on Hyperparameter Settings for Spectral Cluster Number $K_c$ and Spatial Cluster Number $K_s$:** As shown in Table 13 and Table 14, we discussed the performance when fixing the spectral cluster number and observing different spatial cluster numbers, and when fixing the spatial cluster number and observing different spectral cluster numbers, respectively. The reason for choosing fixed $K_c = 3$ and fixed $K_s = 16$ is that extensive experiments have shown that the cluster count of (3, 16) is the most stable and performs best. As shown in Table 13, with fixed $K_c = 3$, as $K_s$ increases, the parameter count and computational volume gradually increase due to the increase in the number of clusters. However, data from groups where $K_s$ is greater than 16 indicates that as $K_s$ becomes larger, performance metrics such as PSNR begin to gradually decrease. This is because as the number of clusters increases, the reconstruction process of spatial pixels is subject to interference from some irrelevant noise centers, resulting in poorer performance. As shown in Table 14, with fixed $K_s = 16$, testing the model's performance under different $K_c$ values reveals that when $K_c$ is greater than 3, as $K_c$ increases, performance metrics like PSNR begin to deteriorate. When the number of clusters equals the number of channels, the model's PSNR decreases by 0.05 dB, and the parameter count increases by 0.02 M. This is because an increase in the number of clusters increases spectral redundancy, and redundant spectra affect the reconstruction quality of the model, thereby reducing model performance.

*Table 13.* Evaluation of Model Performance with Fixed $K_c = 3$ and Varying $K_s$

| $K_s$ | Parameter (M) | FLOPs (G) | PSNR | SSIM | SAM | ERGAS | CC |
|---|---|---|---|---|---|---|---|
| 8 | **1.3128** | **3.0659** | 31.7069 | 0.8391 | 5.5361 | 5.9828 | 0.9019 |
| **16** | 1.3140 | 3.0680 | **31.7262** | **0.8394** | **5.5207** | **5.9788** | **0.9022** |
| 32 | 1.3187 | 3.0763 | 31.7022 | 0.8382 | 5.5444 | 6.0073 | 0.9015 |
| 64 | 1.3373 | 3.0876 | 31.6759 | 0.8393 | 5.5451 | 6.0137 | 0.9013 |
| 256 | 1.7071 | 3.7662 | 31.6035 | 0.8365 | 5.5518 | 6.0455 | 0.9002 |

*Table 14.* Evaluation of Model Performance with Fixed $K_s = 16$ and Varying $K_c$

| $K_c$ | Parameter (M) | FLOPs (G) | PSNR | SSIM | SAM | ERGAS | CC |
|---|---|---|---|---|---|---|---|
| 1 | **1.3140** | **3.0576** | 31.7098 | 0.8390 | 5.5335 | 5.9834 | 0.9022 |
| **3** | 1.3140 | 3.0680 | **31.7262** | **0.8394** | **5.5207** | **5.9788** | **0.9022** |
| 5 | 1.3140 | 3.0711 | 31.7140 | 0.8391 | 5.5298 | 5.9864 | 0.9019 |
| 8 | 1.3141 | 3.0720 | 31.6931 | 0.8380 | 5.5450 | 6.0000 | 0.9015 |
| 102 | 1.3350 | 3.1422 | 31.6737 | 0.8382 | 5.5424 | 6.0308 | 0.9014 |

## I. SR-HSI Classification Experiments

To further verify the proposed method, we conducted classification experiments on the results of the ×2 scaling super-resolution task on the Pavia Center dataset. We used the Iterative self-organizing data analysis technique algorithm (ISODATA) in ENVI software to classify the targets of results from different methods. The number of classification categories was set to 5, and the maximum number of iterations was 100. The visualization results are shown in Figure

13. Our classification results are closest to the ground truth (GT) labels, indicating that this method can more accurately reconstruct high-resolution hyperspectral images with texture and spectral information.

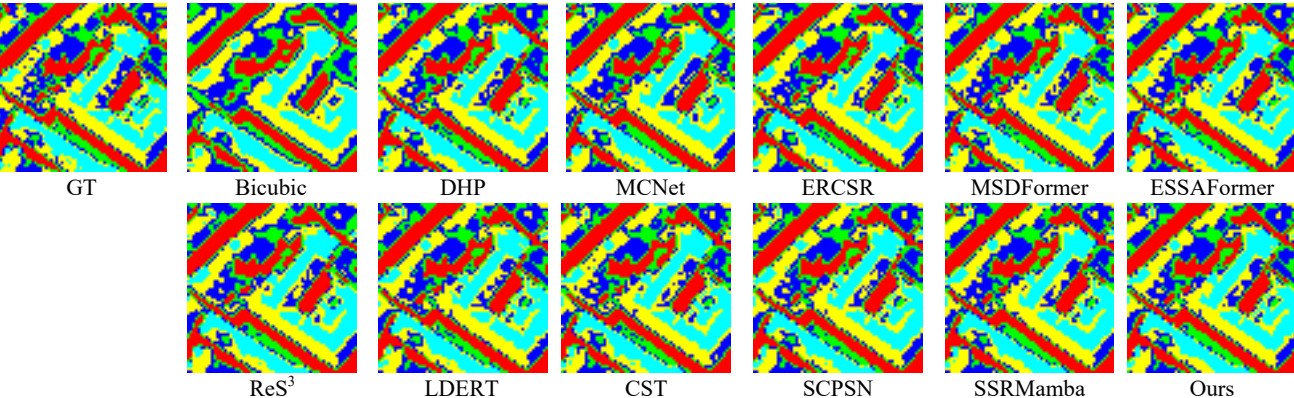

GT  Bicubic  DHP  MCNet  ERCSR  MSDFormer  ESSAFormer

ReS³  LDERT  CST  SCPSN  SSRMamba  Ours

*Figure 13.* Visualization of Classification Results comparing various methods against Ground Truth.

Simultaneously, to more precisely evaluate the classification results, we used two objective evaluation metrics: overall accuracy (OA ↑) and kappa coefficient (Kappa ↑) to assess the quality of the classification results of the SR images. As shown in Table 15, the results indicate that our classification metrics perform the best compared to other methods. This further proves that, compared with existing methods, the method proposed in this paper achieves better reconstruction quality and can significantly improve the accuracy of target classification.

*Table 15.* Comparison of Classification Overall Accuracy (OA) and Kappa Coefficient

| Method | OA | Kappa |
|---|---|---|
| Bicubic | 76.88% | 0.7103 |
| DHP | 87.09% | 0.8377 |
| MCNet | 85.18% | 0.8139 |
| ERCSR | 85.82% | 0.8217 |
| MSDFormer | 87.18% | 0.8390 |
| ESSA | 87.77% | 0.8462 |
| CST | 86.96% | 0.8362 |
| SCPSN | 87.50% | 0.8429 |
| Ress | 85.52% | 0.8181 |
| SSRmamba | 87.28% | 0.8402 |
| LDERT | 86.62% | 0.8318 |
| **Ours** | **88.55%** | **0.8561** |

