# OpenReview forum: "SSDCN: Spatial-Spectral Dual-Clustering-based Network for Hyperspectral Image Super-resolution"
_ICML.cc/2026/Conference — ICML 2026 regular_

### Official Review · Reviewer_uaTP · 2026-03-09

**Soundness:** 3
**Presentation:** 3
**Significance:** 3
**Originality:** 3
**Overall Recommendation:** 5
**Confidence:** 5

**Summary:**

This paper proposes the Spatial-Spectral Dual-Clustering-based Network (SSDCN) for Hyperspectral Image Single Image Super-Resolution (HSI-SISR). To address the quadratic complexity of standard Transformers and the limited receptive fields of window-based methods, the authors introduce the Spatial-Spectral Dual-Cluster Block (SSDCB). This module uses content-driven K-Means clustering to learn low-rank structural bases, achieving global spatial-spectral modeling with linear complexity "O(KN)" . Furthermore, the paper designs a pyramid progressive architecture equipped with a Feature Reuse Reconstruction Block (FRRB), which minimizes computational redundancy by reusing the core tensor and spectral factors from previous scales. The authors also introduce a Pyramid Hierarchical Reconstruction Joint Loss to supervise intermediate features. Extensive experiments on the Pavia Center, Botswana, and Washington DC Mall datasets demonstrate that SSDCN achieves state-of-the-art (SOTA) performance with significantly fewer parameters (1.31M) and FLOPs (3.07G).

**Compliance With Llm Reviewing Policy:**

Affirmed.

**Final Justification:**

The authors have addressed all my concerns, therefore I decide to raise the score to 5.

**Key Questions For Authors:**

(1) Since K-Means is inherently a non-differentiable hard clustering algorithm, how does the SSDCB address the gradient interruption to ensure stable end-to-end training?
(2) The clustering mechanism in SSDCB relies on spectral similarity. How does the model maintain performance when dealing with severe or non-uniform noise common in real-world HSIs?
(3) Regarding the selection of $K_c$ and $K_s$, what are the underlying principles for balancing representation capability and computational efficiency? Are there general guidelines for adapting these parameters to datasets with varying spectral bands or spatial complexities?

**Limitations:**

No. The authors have not explicitly discussed the limitations or potential negative societal impact of their work. The manuscript could be further strengthened by adding a concise section regarding its limitations and potential societal impacts.

**Strengths And Weaknesses:**

The paper proposes a dual-clustering mechanism as an efficient alternative to point-to-point attention, leveraging the spectral low-rank and spatial self-similarity properties of HSIs. The network design, particularly the FRRB module that utilizes hierarchical invariance to reduce redundant decompositions, is logically sound and supported by comprehensive ablation studies and downstream classification tasks. By achieving competitive PSNR performance with significantly fewer parameters and FLOPs than models like LDERT and ESSAFormer, the work offers a practical trade-off for resource-constrained HSI super-resolution. While the manuscript is well-structured and the visual diagrams (e.g., Figures 2 and 6) effectively clarify the methodology, the robustness of the K-means clustering process to severe environmental or sensor noise remains a point that warrants further discussion.

---

> ### Author Rebuttal · Authors · 2026-03-30
>
> ### Reviewer #3 (uatp)
>
> **1. Response to Weaknesses & Question 2:**
> Regarding clustering based on spectral similarity matrices and direct spectral clustering, the actual runtime is found to be comparable through testing. Taking quadruple super-resolution as an example, the average runtime is 0.13 seconds for both methods. However, due to the computational duration of the spectral similarity matrix calculation, the overall runtime of the former is extended by 0.09 seconds.While our overall computational load is marginally increased compared to direct clustering, this spectral similarity matrix is absolutely not redundant. It is deliberately designed to robustly resist noise. K-means is highly sensitive to noise; slight variations drastically alter cluster centers, degrading assignments. Hyperspectral images inherently suffer from stripe and random noise. Clustering directly on the HSI allows this noise to severely interfere with classification, degrading super-resolution reconstruction. By clustering the spectral similarity matrix, we elegantly circumvent this issue because noise is localized and its impact on global overall similarity is minimal. The final clustering result acts as a weighted sum of the similarity centers for each spectrum, providing an excellent smoothing effect for spatial noise. Consequently, during subsequent spatial dimensionality reductions, we can proceed confidently without worrying about noise interference.
>
> Table 4: Experimental comparison of LRHSI with slight random noise added on Pavia Dataset
>
> |Input|PSNR|SSIM|SAM|ERGAS|RMSE|CC|
> |:---|:---|:---|:---|:---|:---|:---|
> |SSDCN with Spectral Similarity Matrix Clustering |31.5917|0.8312|5.7051|6.0704|0.0322|0.8990|
> |SSDCN Direct Spectral Clustering |31.1858|0.8157|5.9000|6.3179|0.0336|0.8905|
>
> **2. Response to Question 1:**
> As illustrated in Figures 1 to 3, our clustering method exists exclusively within the SSDCB module; there are zero clustering operations within the FRRB. SSDCB sits at the extreme front of the entire model pipeline, directly receiving the LRHSI. Inside SSDCB, our two clustering modules are placed foremost. Crucially, all deep learning modules requiring parameter updates during training are located strictly after the clustering operations. Because no learnable modules precede clustering, there is no requirement for gradient backpropagation through the clustering step itself; gradients only backpropagate up to the modules immediately following it. This structural isolation makes implementation exceptionally flexible. Our comparisons proved K-means hard clustering performs better than soft clustering. It can be constructed natively via PyTorch, or one could call Sklearn's KMeans to execute CPU clustering before GPU training, without affecting results. For our FRRB ablation, residual connections forced gradient backpropagation before clustering. Table 6 proves our strategy is highly effective: using SSDCB to replace FRRB still outperforms SOTAs like SCPSN. Our FRRB design uniquely accelerates computation by bypassing redundant clustering while ensuring smooth gradient flow. Our model adopts PyTorch-based K-Means hard clustering, converging stably within five iterations.
>
> **3. Response to Question 3:**
> We discussed hyperparameter impacts in Appendix Tables 11 and 12. The PaviaX4 experiment clearly demonstrates that setting $(K_c, K_s) = (3, 16)$ achieves the optimal balance among total parameter count, computational load (FLOPs), and reconstruction accuracy (PSNR). Why is performance best when $K_c=3$? HSI data possesses extremely high spectral redundancy (as shown in Figure 5, inter-band correlation is strong). Increasing $K_c$ beyond 3 introduces additional basis vectors containing redundant information or disruptive noise, which conversely leads to a noticeable decline in PSNR (Table 12). This proves a small number of spectral bases is entirely sufficient to correctly characterize the intrinsic low-rank properties of HSI. Why is the effect best when $K_s=16$? Spatial clustering inherently aims to accurately capture non-local self-similarity. When $K_s < 16$, the model cannot fully express complex textures. Conversely, when $K_s > 16$, too many cluster centers introduce irrelevant, disruptive noise centers that actively interfere with pixel reconstruction, leading to performance degradation (Table 11). For absolute fairness, we adopted the $(3, 16)$ pair for all primary experiments, successfully achieving SOTA results (except for ERGAS and CC on Washington DC X8). For entirely different datasets, optimal parameters might vary. Our suggested general principle is: $K_c$ should firmly align with the intrinsic rank of the HSI (usually 3-5 is sufficient), while $K_s$ should directly correlate with the texture complexity of the specific scene. For highly complex textures, $K_s$ can be appropriately increased, but caution is necessary to definitively prevent overfitting.

---

> > ### Author Rebuttal · Reviewer_uaTP · 2026-04-03
> >
> > The authors have addressed all my concerns, therefore I decide to raise the score to 5.

---

> > > ### Author Response · Authors · 2026-04-03
> > >
> > > We really appreciate the reviewer’s recognition of our work. We will carefully revise and improve the manuscript according to the comments after the manuscript is accepted.

---

### Official Review · Reviewer_8BEY · 2026-03-12

**Soundness:** 2
**Presentation:** 3
**Significance:** 2
**Originality:** 3
**Overall Recommendation:** 4
**Confidence:** 5

**Summary:**

This paper proposes SSDCN, a hyperspectral image single image super-resolution (HSI-SISR) network that replaces standard self-attention with a spatial-spectral dual-clustering mechanism to achieve linear complexity. The method has three main components: (1) a Spatial-Spectral Dual-Cluster Block (SSDCB) that performs sequential spectral and spatial K-Means clustering on feature matrices to derive a core tensor G, spectral factor matrix U_spe, and spatial factor matrix U_spa, followed by a Factor Matrix Nonlinear Correction Unit (FMNCU) to map these to SR-capable representations; (2) a Feature Reuse Reconstruction Block (FRRB) for higher scale levels (×2, ×4) that reuses the core tensor and spectral factor matrix from the ×1 level and only recomputes the spatial factor; and (3) a pyramid hierarchical reconstruction joint loss that supervises intermediate scale outputs.

**Compliance With Llm Reviewing Policy:**

Affirmed.

**Final Justification:**

My questions are all addressed.

**Key Questions For Authors:**

(1) How exactly are gradients propagated through the K-Means clustering operations (Eqs. 2–6) during training? Are the cluster assignments treated as fixed (computed in a detached forward pass) with only the FMNCU and convolutional layers being optimized, or is some differentiable approximation (e.g., soft K-Means, Gumbel-Softmax) employed?

(2) The paper claims SSDCB achieves "global modeling" , but the K-Means step compresses all spatial information into K_s cluster centers, which are then used identically for all pixels via a single matrix multiplication. How does this fundamentally differ from a low-rank projection or global average pooling in terms of its capacity to model position-dependent, non-local spatial relationships?

(3) On the Washington DC Mall dataset at ×8 of Table 3, SSDCN's ERGAS is notably worse than SCPSN, ReS³, and SSRMamba, and its CC is lower than SSRMamba and SCPSN. Because this is the highest scale factor where the method's assumptions about spectral invariance are most stressed, does this suggest that reusing the ×1 core tensor and spectral factors breaks down for large scale factors, and would re-running SSDCB at each level improve results?

**Limitations:**

While SSDCN demonstrates strong efficiency-performance trade-offs, several limitations should be noted. First, the paper does not explain how gradients propagate through the non-differentiable K-Means clustering operations during backpropagation, leaving the end-to-end trainability of the core tensor and factor matrices theoretically unclear. Second, the claimed linear complexity O(KN) does not account for the iterative cost of K-Means itself, and actual wall-clock inference times are not reported, making the practical efficiency advantage difficult to verify beyond FLOPs counts. Fifth, experiments are conducted on only three relatively small hyperspectral benchmarks with a fixed 3:1 train-test split and no reported confidence intervals or statistical significance tests, which limits the generalizability of the conclusions.

**Strengths And Weaknesses:**

**Strengths**
1. The paper addresses a conffict in HSI-SISR between global non-local modeling and computational efficiency. The proposed clustering-based decomposition offers a new alternative to the window-attention paradigm that dominates the field.
2. The model complexity of this method is superior. At 1.31M parameters and 3.07G FLOPs, SSDCN is an order of magnitude lighter than most competitors while achieving the highest PSNR on Pavia Center ×4.
3. The dual-clustering ablation in Table 5 systematically validates that both spatial and spectral dimensionality reduction contribute to performance, with the full model achieving the best metrics at the lowest FLOPs.

**Weaknesses**
1. K-Means clustering is non-differentiable, yet the paper never addresses how gradients flow through the clustering operations in Eqs. 2, 3, 5, and 6 during backpropagation. K-Means involves discrete assignment steps (argmin over cluster indices) that have zero gradients almost everywhere, meaning the core tensor G and the cluster-derived factor matrices cannot be optimized end-to-end through gradient descent in the standard sense. The paper describes SSDCB as if these operations are seamlessly integrated into the training pipeline, but without explaining the gradient approximation or straight-through estimation strategy employed, the claimed **content-driven** learning of low-rank bases is theoretically unsubstantiated. Maybe the network is simply learning around fixed clustering assignments.
2. The "linear complexity O(KN)" claim (Abstract, Sec. 1) is misleading: K-Means itself has complexity O(NKI) per iteration where I is the number of iterations, and the paper does not specify how many K-Means iterations are run per forward pass or whether convergence is guaranteed within the training loop; the actual wall-clock inference time is also never reported, despite being the most relevant efficiency metric.
3. The experimental protocol uses only three small datasets with a fixed 3:1 train-test split, no cross-validation, and no error bars or statistical significance tests, making it difficult to assess whether the improvements are meaningful or within noise.

---

> ### Author Rebuttal · Authors · 2026-03-30
>
> ### Reviewer #2 (8BEY)
>
> **1. Response to Weaknesses 1 & 2 & Question 1:**
> As shown in Figures 1-3, the clustering process occurs exclusively at the front end of the SSDCB module when receiving the LRHSI. All learnable deep learning modules strictly follow the clustering operation. Because no learnable parameters precede clustering, gradient backpropagation completely stops before this step. This isolation mechanism enables highly flexible deployments. The hard K-means clustering algorithm (converging within 5 iterations via PyTorch) outperforms soft clustering, and can even execute on CPU via Sklearn without affecting gradient computations. For the FRRB ablation study where SSDCB replaces FRRB, we implemented residual connections before clustering to allow gradients to backpropagate. Table 6 demonstrates that replacing FRRB with SSDCB significantly improves performance, outperforming SOTA methods like SCPSN. Our FRRB design not only accelerates computation speed but also ensures stable gradient flow. Crucially, the claimed "linear complexity" strictly refers to capturing global spatial information ($O(K_c \times N)$) and spectral information ($O(K_s \times C)$), rather than the K-means algorithm or spectral similarity matrix complexity. Regarding "low-rank basis learning," it is a structured representation learning paradigm for high-dimensional redundant data, adaptively learning a minimal set of linearly independent low-rank basis vectors. Our model innovatively uses clustering initialization followed by deep learning network training to solve for the low-rank basis, rather than learning it peripherally.
> | Model | Time/s | Top Memory/GB |
> | :--- | :--- | :--- |
> | ESSAFormer(2023-ICCV) | 0.5702 | 0.3228 |
> | CST(2024-TIP) | 0.6914 | 0.0653 |
> | SCPSN(2024-ACMMM) | 0.6242 | 0.0378 |
> | LDERT(2025-TPAMI) | 0.6782 | 0.7968 |
> | SSDCN(Ours) | 0.5139 | 0.0293 |
>
> **2. Response to Weaknesses 3:**
> We acknowledge omitting error bars. We supplemented experiments using different random seeds on PaviaX4. Our metrics remain highly stable (PSNR consistently > 31.70, SAM tightly bounded $\le$ 5.61), formally confirming our massive advantage over SOTA methods.
>
> | Model | PSNR | SSIM | SAM | ERGAS | RMSE | CC |
> |:---|:---|:---| :--- | :--- | :--- | :--- |
> | SSDCN(seed=10) | 31.7027 | 0.8393 | 5.5346 | 5.9959 | 0.0316 | 0.9017 |
> | SSDCN(seed=20) | 31.7344 | 0.8396 | 5.5551 | 5.9748 | 0.0316 | 0.9023 |
> | SSDCN(seed=30) | 31.7083 | 0.8385 | 5.6143 | 5.9791 | 0.0317 | 0.9014 |
> | SSDCN(seed=40) | 31.7241 | 0.8391 | 5.5302 | 5.9857 | 0.0316 | 0.9022 |
> | SSDCN(seed=50) | 31.7337 | 0.8396 | 5.5850 | 5.9756 | 0.0316 | 0.9023 |
> | SSDCN(Original) | 31.7262 | 0.8394 | 5.5207 | 5.9788 | 0.0316 | 0.9022 |
>
> **3.Response to Question 2:**
> Global Average Pooling unconditionally compresses pixels into one global vector, losing location-dependent texture differences and discriminability. In Appendix Table 9, replacing clustering with adaptive pooling reduced performance because rigidly fixed windows cannot represent diverse semantic categories within complex images. Low-Rank Projection reduces dimensionality via a static learnable matrix. Conversely, our method dynamically generates the initial factor matrix by multiplying all pixels with real-time cluster centers, uniquely adapting to each image. It is inherently content-aware. Substituting clustering with a 1x1 convolution (Appendix Table 9) noticeably degraded performance. Furthermore, projection merely yields the initial factor matrix; our FMNCU actively learns and mines complex non-local spatial relationships hidden deeply within it.
>
> **4.Response to Question 3:**
> Spectral invariance is an established theory, not our hypothesis. "EigenSR" (AAAI 2025) rigorously proved it, enabling RGB to HSI transfer learning. Classical spectral unmixing mathematically supports this: an HSI ($C \times H \times W$) decomposes into endmembers ($C \times K$) and abundance maps ($K \times H \times W$). Applying super-resolution to spatial abundance maps and subsequently multiplying by original endmembers reconstructs the HSI. Throughout this process, endmembers fundamentally remain constant. Our spectral factor matrix is mathematically equivalent to these endmembers. The primary working hypothesis of our paper is core tensor invariance. Ablation testing on the FRRB reuse mechanism showed repeatedly decomposing hierarchical levels directly reduces performance while increasing computational load. We added an ablation (Washington DC, X8) confirming repeated decomposition (SSDCB replacing FRRB) degrades results. ERGAS and CC measure spatial/spectral fidelity; for the universally recognized PSNR and SAM metrics representing those exact qualities, our method achieves absolute SOTA results.
>
> | Model | PSNR | SSIM | SAM | ERGAS | RMSE | CC |
> |:---|:---|:---|:---|:---|:---|:---|
> | SSDCB Replace FRRB | 42.7136 | 0.8383 | 8.7691 | 11.159 | 0.0243 | 0.6817 |
> | SSDCN | 42.8261 | 0.8439 | 8.1663 | 10.9594 | 0.0242 | 0.6849 |

---

> > ### Author Rebuttal · Reviewer_8BEY · 2026-04-02
> >
> > My questions are all addressed. The score can be improved to 4.

---

> > > ### Author Response · Authors · 2026-04-02
> > >
> > > We really appreciate the reviewer’s recognition of our work. We will carefully revise and improve the manuscript according to the comments after the manuscript is accepted.

---

### Official Review · Reviewer_gg3p · 2026-03-13

**Soundness:** 2
**Presentation:** 3
**Significance:** 2
**Originality:** 2
**Overall Recommendation:** 2
**Confidence:** 4

**Summary:**

This paper proposes SSDCN for single-image hyperspectral image super-resolution using a progressive pyramidal pipeline. At the base level, it employs the Spatial-Spectral Dual-Clustering Block (SSDCB) to perform spectral and spatial clustering, forming a core tensor and factor matrices for factorized reconstruction. At higher levels, the Feature Reuse and Refinement Block (FRRB) reuses the core tensor and spectral factors from the base level and updates the spatial factors across scales, with High-Resolution Basic Blocks (HRBS) applied in a residual manner. The Factor Matrices Nonlinear Correction Unit (FMNCU) is used to adjust the factor matrices, and training includes a hierarchical multi-scale loss to supervise intermediate outputs.

**Compliance With Llm Reviewing Policy:**

Affirmed.

**Final Justification:**

After reading the responses, my overall assessment remains unchanged. The main reason is that the complexity concern is still not fully resolved. In particular, the current explanation narrows the linear-complexity claim to the global interaction step, whereas the manuscript presents SSDCB/SSDCN more broadly as achieving global modeling with linear complexity. Moreover, the rebuttal’s complexity accounting for the spectral similarity computation still appears insufficiently rigorous relative to the formulation in the paper. The novelty and significance concerns also remain, since the new response mainly reframes the contribution at a conceptual level but does not provide sufficiently strong new evidence to establish a substantial methodological advance over prior clustering-based or efficiency-oriented designs. The clarification on code availability is helpful and partially improves reproducibility, but it is not sufficient to change my final recommendation. Therefore, my final recommendation remains

**Key Questions For Authors:**

1. Given the linear-complexity claim for SSDCB, can the authors provide an end-to-end complexity and runtime breakdown that includes the cost of constructing the spectral similarity matrix and running clustering iterations?

2. What are the practical inference latency and peak memory usage of SSDCN for representative band counts and spatial resolutions, and how do these compare to the strongest baselines under the same hardware and implementation setting?

3. How exactly is K-Means implemented in SSDCB and FRRB, including initialization, iteration count or stopping criterion, whether clustering is recomputed or cached, and how are non-differentiable assignments handled during training?

4. Why does the training protocol switch batch size from 2 to 30 after 5000 epochs, and do the main results remain consistent under a standard fixed-batch schedule with comparable training budgets for baselines?

5. Can the authors include a tightly controlled baseline that replaces SSDCB with a learned low-rank or lightweight global-interaction module under matched compute and training budget within the same pyramidal pipeline, to test whether clustering is necessary for the reported gains?

**Limitations:**

The paper does not provide end-to-end runtime or memory results, nor a complete cost breakdown including similarity construction and clustering iterations, so the efficiency claim is difficult to verify. Key K-Means and training-protocol details are underspecified, limiting reproducibility. The reported improvements may be sensitive to training choices and do not clearly establish a substantial methodological advance.

**Strengths And Weaknesses:**

Strengths:
SSDCB introduces spatial–spectral dual clustering to build a factorized reconstruction with an explicit core tensor and factor matrices.
FRRB reuses the base-level core tensor and spectral factors at higher scales to limit repeated computation.


Weaknesses:
Marginal Novelty. SSDCN largely combines a clustering-based factorized reconstruction module (SSDCB) with a conventional progressive pyramidal super-resolution pipeline, multi-scale supervision, and a cross-scale reuse mechanism (FRRB) that mainly serves as an implementation-level efficiency refinement. Overall, the contribution appears incremental, lacking fundamental methodological or theoretical breakthroughs relative to existing factorization- and efficiency-oriented designs.

Unsubstantiated Complexity Advantage. The paper repeatedly claims that SSDCB enables global modeling with linear complexity, but the method explicitly forms a spectral similarity matrix, which incurs computation and memory costs that scale quadratically with the number of bands and are not reconciled with the linear-complexity narrative. In addition, the efficiency evidence mainly reports parameters and FLOPs, without end-to-end runtime results or a cost breakdown that accounts for similarity construction and clustering iterations, so the claimed speed advantage is difficult to verify.

Underspecified Clustering and Training Procedure. SSDCB uses K-Means for spectral clustering and for spatial clustering to obtain the core tensor used in reconstruction. The paper does not report key implementation details for K-Means, including initialization, iteration count, whether clustering is recomputed or cached, and how non-differentiable assignments are handled, which limits reproducibility. It also adopts a two-stage training schedule that switches batch size from 2 to 30 after 5000 epochs, without a clear rationale.

---

> ### Author Rebuttal · Authors · 2026-03-30
>
> ### Reviewer #1 (gg3p)
>
> **1. Response to Weaknesses 1:**
> Unlike current SOTA methods (ESSAFormer, CST, LDERT, SCPSN) that optimize computational efficiency by reducing global attention complexity, our method introduces an original global dependency modeling paradigm based on spatial-spectral dual clustering. It is not an improvement on attention but a novel construction of factor matrices using the relationship between global pixels/spectra and cluster centers. This represents a high scale of innovation compared to existing module stacking approaches.
>
> **2. Response to Weaknesses 2 & Question 1 & Question 2:**
> In HSI-SISR, "quadratic complexity" typically refers to the spatial domain ($O(N^2)$). SOTA methods (ESSAFormer, CST, LDERT) advertise "linear complexity" by making spatial computations linear, despite retaining quadratic spectral complexity ($O(C^2)$). Similarly, our method strictly achieves linear complexity for modeling factor matrices: $O(K_s \times N)$ spatially and $O(K_c \times C)$ spectrally.
>
> Our reported GFLOPs include the clustering overhead. Regarding clustering based on spectral similarity matrices and direct spectral clustering, the actual runtime is found to be comparable through testing. Taking 4×SR task as an example, the average runtime is 0.13 seconds for both methods. However, due to the computational duration of the spectral similarity matrix calculation, the overall runtime of the former is extended by 0.09 seconds.However, this design aims to provide the model with critical noise resistance capabilities. The K-means algorithm exhibits high sensitivity to high-frequency noise. By utilizing the global similarity matrix of clustering, local noise can be effectively smoothed, preventing significant shifts in cluster centers that may degrade reconstruction quality.
>
> Table 1: End-to-end time and peak memory comparison with Transformer-based models (for 4x SR tasks with input size (1,102,16,16))
>
> |Model|Time/s|Top Memory/GB|
> |:---|:---|:---|
> |ESSAFormer(2023-ICCV)|0.5702|0.3228|
> |CST(2024-TIP)|0.6914|0.0653|
> |SCPSN(2024-ACMMM) |0.6242|0.0378|
> |LDERT(2025-TPAMI) |0.6782|0.7968|
> |SSDCN(Ours) |0.5139|0.0293|
>
> Table 2: Breakdown analysis of end-to-end operational complexity
>
> |SSDCN Modules|Time/s|
> |:---|:---|
> |Channel Similarity Matrix Compute|0.0960|
> |Channel Clustering|0.1349|
> |Spa Clustering|0.0432|
> |Spe FMNCU|0.0354|
> |Spa FMNCU|0.0012|
> |Reconstruct|0.0029|
> |FRRB|0.1051|
>
> Table 4: Experimental comparison of LRHSI with slight random noise added on Pavia Dataset
>
> |Input|PSNR|SSIM|SAM|ERGAS|RMSE|CC|
> |:---|:---|:---|:---|:---|:---|:---|
> |SSDCN with Spectral Similarity Matrix Clustering|31.5917|0.8312|5.7051|6.0704|0.0322|0.8990|
> |SSDCN Direct Spectral Clustering|31.1858|0.8157|5.9000|6.3179|0.0336|0.8905|
>
> **3. Response to Weaknesses 3 & Question 3 & Question 4:**
> As shown in Figures 1-3, clustering exists exclusively at the very front of the SSDCB module, preceding all learnable deep learning components. Therefore, gradient backpropagation absolutely does not need to pass through the clustering operation itself; gradients only flow back to the modules immediately following it. This isolation ensures stable gradient flow and enables flexible implementations. We utilize PyTorch-based hard K-Means, which consistently converges within 5 iterations and outperforms soft clustering. For the FRRB ablation study where SSDCB replaces FRRB, we through residual connection to allow gradients to backpropagate. Table 6 demonstrates that replacing FRRB with SSDCB significantly improves performance, outperforming SOTA methods like SCPSN. Our FRRB design not only accelerates computation speed but also ensures stable gradient flow.Finally, our two-stage training strategy—initially using a small batch with a higher learning rate to escape local minima, followed by a large batch with a small learning rate for stable convergence—significantly outperforms StepLR and Cosine Annealing.
>
> |Training Strategy|PSNR|SSIM|SAM|ERGAS|RMSE|CC|
> |:---|:---|:---|:---|:---|:---|:---|
> |StepLR halves learning rate every 3000 epochs|31.6926|0.8380|5.5736|5.9832|0.0316|0.9021|
> |Cosine Annealing|31.6831|0.8376|5.5623|5.9964|0.0317|0.9010|
> |Two-stage training (Ours)|31.7262|0.8394|5.5207|5.9788|0.0316|0.9022|
>
> **4. Response to Question 5:**
> Because FRRB relies on SSDCB's feature extraction, substituting SSDCB requires substituting FRRB. Replacing them with popular attention mechanisms (ESSA or Rectangle) degraded performance. In Table 9 of the main text, replacing the clustering step with convolutional layers or adaptive pooling also resulted in performance drops, confirming our model's gains depend heavily on the clustering mechanism.
>
> |Model|PSNR|SSIM|SAM|ERGAS|RMSE|CC|
> |:---|:---|:---|:---|:---|:---|:---|
> |Replace SSDCB/FRRB with ESSA Attention|31.4925|0.8331|5.7180|6.1209|0.0322|0.8971|
> |Replace SSDCB/FRRB with Rectangle Attention|31.5537|0.8352|5.5719|6.0752|0.0322|0.8979|
> |Ours|31.7262|0.8394|5.5207|5.9788|0.0316|0.9022|

---

> > ### Author Rebuttal · Reviewer_gg3p · 2026-04-04
> >
> > Thank you for the detailed rebuttal. The additional runtime and memory results, training ablations, and clarifications regarding K-Means are helpful and improve the paper’s reproducibility. However, my final recommendation remains Reject. The main reason is that the rebuttal does not fully address my core concerns regarding novelty, significance, and the complexity claim. In the manuscript, SSDCB/SSDCN is repeatedly presented as achieving global modeling with linear complexity, while the method explicitly constructs a spectral similarity matrix and performs clustering. The current response supports practical efficiency rather than a fully self-consistent linear-complexity narrative. Although the new evidence is useful, it still does not convincingly establish a substantial methodological advance over prior clustering-based or efficiency-oriented designs, nor does it fully eliminate the reproducibility concerns in the original submission.

---

> > > ### Author Response · Authors · 2026-04-05
> > >
> > > (1) Complexity Claim:
> > >
> > > 1.First, we must clarify that our statement regarding the linear complexity of global modeling strictly follows the recognized academic consensus in efficient Transformers and vision modeling. There is no contradiction. In this field, the extensively discussed quadratic complexity specifically refers to the $O(N^2)$ bottleneck of token-to-token global interaction, while the overall self-attention calculation is $O(N^2C)$ (where $N$ is spatial pixels/tokens, $C$ is channel dimension).
> > >
> > > 2.Recent top-tier papers in this domain do not include pre-interaction operations in their global complexity claims. In HSI-SISR, optimizations targeting quadratic complexity specifically refer to optimizing the $O(N^2)$ global interaction. This includes the rectangular window attention in CST (Chen et al., 2024) and LDERT (Li et al.,2025b), and the kernelized attention in ESSA(Zhang et al., 2023). Their claimed "linear complexity" strictly targets the global interaction step.
> > >
> > > 3.Our model's global interaction involves $K$ cluster centers and $N$ spatial pixels, yielding a complexity of $O(KN)$. Your concern stems from the preceding steps: solving the similarity matrix ($O(C^2)$) and K-Means clustering ($O(tkNC)$). Since $t$ and $k$ are fixed smaller hyperparameters and $N \gg C$, our clustering step simplifies to $O(NC)$. Importantly, deriving the initial Q, K, and V tokens in standard self-attention before any interaction also requires $O(NC)$ time.
> > >
> > > 4.If our linear complexity claim is invalidated due to pre-interaction computations, then none of the published SOTAs can be considered "linear" either, as their preliminary operations are on the exact same magnitude. Furthermore, our empirical comparisons show that among models with linear self-attention optimizations, our model achieves the lowest actual runtime.
> > >
> > > (2) Novelty & Significance:
> > >
> > > Regarding the methodological advancement over previous clustering designs, we emphasize a fundamental paradigm shift. We fully acknowledge that clustering has been applied in HSI-SISR. In fact, SCPSN (Yang et al., 2024) successfully used spectral clustering to aggregate super-channels and reconstruct high-resolution images via a low-rank subspace. However, SCPSN and almost all traditional HSI-SISR models remain strictly confined to the conventional paradigm of learning direct pixel-to-pixel relationships.
> > >
> > > We introduce a completely new learning paradigm: shifting from learning local pixel mappings to learning global factor matrix representations. SSDCN no longer forces the network to learn exhaustively redundant local mappings between pixels. The initial clustering extracts $K$ spectral and spatial centers as explicit global context representations. Interacting these centers with all pixels constructs a highly informative factor matrix. The deep learning units within SSDCB/FRRB are exclusively tasked with reconstructing and optimizing these latent factor matrix features, avoiding the dense calculation of pixel pairs. This profoundly reimagines how neural networks efficiently utilize global hyperspectral information, fundamentally changing super-resolution mechanics and achieving a true methodological breakthrough.
> > >
> > > (3) Reproducibility Concerns:
> > >
> > > We promise that upon acceptance, we will open-source the complete training and inference code, pre-trained weights, and detailed experimental configurations to ensure full reproducibility. In fact, we have already uploaded the code in our supplementary materials.
> > >
> > > (4) Summary:
> > >
> > > In summary, our linear complexity claim is mathematically rigorous and perfectly aligns with standard terminology. Our use of clustering grounds a novel factor-matrix low-rank basis reconstruction paradigm that structurally departs from traditional pixel-mapping architectures. We firmly believe this work makes an important contribution to the field. Given these detailed clarifications, we respectfully request that you reconsider the novelty and rigor of our proposed paradigm. Thank you again for your insightful comments.

---

### Decision · Program_Chairs · 2026-04-30

**Decision:**

Accept (regular)

**Comment:**

This paper proposes a hyperspectral image super-resolution method using a spatial-spectral dual-clustering mechanism to reduce the computational load. All three reviewers noted the insufficient explanation of the K-means step and raised questions regarding implementation details, including:

1. Why the authors claim the proposed method has a complexity of $O(KN)$ when K-means clustering’s complexity is $O(KNI)$, where $I$ is the number of iterations.

2. The error bars for the K-means clustering’s random initialization.

3. How derivatives are backpropagated through the non-differentiable K-means clustering.

4. The inference latency for individual steps.

5. The reasoning behind the proposed training protocol.

The authors provided detailed explanations and additional quantitative data to address these questions. All three reviewers acknowledged that the additional explanations and numerical analysis were helpful. Ultimately, one reviewer still had concerns about the novelty and gave a "Reject" recommendation. In contrast, the other two reviewers acknowledged that their concerns were fully resolved and raised their recommendations to "Weak Accept" and "Accept" with the highest confidence scores.

The authors successfully provided the missing details in their rebuttals, convincing the reviewers of the technical soundness of the proposed method. While the level of novelty remains somewhat subjective, the overall quality of the work justifies its inclusion.

Putting everything into consideration, the paper is recommended for acceptance. The promising improvements, clear presentation, and thorough disclosure of experimental settings and implementation details are highly commendable. It is expected that the authors will incorporate the reviewers’ feedback to further enhance clarity and maximize the paper's contribution to the community.